# Social avoidance can be quantified as navigation in abstract social space
Matthew Schafer [1,2] ✉ & Daniela Schiller [1,3,4] ✉

We navigate social relationships daily, making decisions that can change our affiliation and power relations with others. People high in social avoidance report perceiving little affiliation and power in their social lives. Do they also make low affiliation and low power interaction choices in actual social interactions? We hypothesized that social avoidance can be quantified as navigation in an abstract social space framed by power and affiliation. To test this, we recruited two large samples of online participants (Initial sample $n = 579$, Validation sample $n = 255$) to complete a naturalistic social interaction game where they form relationships with fictional characters, and a battery of questionnaires. Factor analysis of the questionnaires revealed a social avoidance factor that related to a low affiliation and low power interaction style, which was reflected in large social distance between the participants and characters. This distance, in turn, was related to smaller and less complex real-world social networks—suggesting that this abstract behavioral geometry reflects real-life behavioral tendencies. Language analysis of post-task character descriptions found semantic representations that mirrored the relationships formed in the task, with social avoidance relating to more negative impressions. This approach suggests that social avoidance can be thought of as an abstract, two-dimensional navigational strategy, potentially leading to effective strategies for social skills training and therapy.

Affiliation and power are fundamental in social behavior[1], underlying stereotyping (warmth and competence[2]), facial impressions (trustworthiness and dominance[3]) interpersonal traits and behaviors[4], as well as relationships of non-human primates (bonding and dominance[5]) and other animals. These social dimensions manifest differently across people, with individuals high in social avoidance consistently self-reporting feeling low affiliation with others and feelings of powerlessness. For example, they report expecting social rejection[6], behaving submissively[7], and lacking motivation to pursue affiliative interactions[8]. However, some work suggests social avoidance is related to power, but not affiliation[9], and it is generally unclear if and how these self-reports relate to actual behaviors in everyday social settings. Quantifying socially avoidant behavior during ongoing social encounters could provide a complementary tool to self-reports, and a way to implicitly track social avoidance tendencies, to reshape behavior and track treatment efficacy. Thus, here we sought to test the extent to which people high in self-reported social avoidance show implicit low affiliation and low power behavioral tendencies in social interactions, using a behavioral paradigm designed to capture naturalistic social behavior—the social navigation task.

We view social avoidance as an abstract navigational strategy: individuals minimize the negative consequences of social interactions by avoiding affiliation and acting submissively, positioning themselves in a low affiliation and low power state.

Evidence from the social navigation task supports this view. In this task, social relationships are developed through first-person interaction-based decision-making that changes the relative affiliation and power in relationships, which are represented geometrically as locations in abstract affiliation and power space. Self-reported social avoidance correlates with how the hippocampus—a navigation-related brain region that is also responsive to social anxiety treatment[10]—tracks abstract social relationships in this task[11]. These findings suggest that social avoidance operates as an abstract navigational strategy, shaping the geometry of social space[1,12].

In this study, two large samples ($n = 579$ and $n = 255$) of online participants completed the social navigation task and a series of post-task measures. We expected social avoidance symptoms to have a two-dimensional structure: people highest in these symptoms should make low affiliation and low power decisions in the task's social interactions (**hypothesis 1**); and we expected these behaviors to correlate specifically

[1]Department of Neuroscience, Icahn School of Medicine at Mount Sinai, New York, NY, USA. [2]Department of Psychiatry, Columbia University, New York, NY, USA. [3]Department of Psychiatry, Icahn School of Medicine at Mount Sinai, New York, NY, USA. [4]Friedman Brain Institute, Icahn School of Medicine at Mount Sinai, New York, NY, USA. ✉e-mail: ms6883@columbia.edu; daniela.schiller@mssm.edu

with social avoidance, more so than with other kinds of mental function (**hypothesis 2**). We followed up on these main hypotheses with additional hypotheses in the second sample: we expected people who report higher social avoidance to also have more negative impressions of the characters, and for this to relate to behavior (**hypothesis 3**). Finally, we reasoned that if the behavior in the social task is indicative of real-world social avoidance, then the tendency to create abstract social distance should relate to real-world social network structure (**hypothesis 4**).

To test these hypotheses, we combined geometric measures of social interaction behavior in the social navigation task with a factor analysis on self-report questionnaires, to depict social behaviors as navigational decisions within an abstract space. We also analyzed participant's open-ended text descriptions about the characters using large language models to probe participants' self-reported representations. We found support for all these hypotheses, suggesting that social avoidance has a two-dimensional behavioral structure, which relates to negative self-reported impressions of social interaction partners and smaller real-world social networks.

## Methods
### Independent samples
We collected data from participants on the online platform Prolific. Participants were at least 18 years of age, based in the United States of America with English as their first language, and had been approved on at least 95% of their previous Prolific tasks. We did not know their identities. Participants provided consent by clicking "I Consent" after reading information about the study and were paid for their participation after completion, in accordance with policies on Prolific and at Mount Sinai's Icahn School of Medicine. The study was approved by the ethics committee at Mount Sinai's Icahn School of Medicine.

Data was collected in two independent samples, called the "Initial" and "Validation" samples. To be included, participants had to have a plausible average decision response time in the task (within ±2 standard deviations (SD) of the mean (M)), above-chance post-task memory, and correct responses to attention checks during the questionnaires.

The Initial sample included 579 participants (733 before exclusions, 21% excluded; age: $M = 38.4$, $SD = 13.4$; sex: female participant $n = 288$, male participant $n = 290$, other $n = 1$; race: white $n = 414$, Black $n = 46$, Asian $n = 56$, Native American $= 2$, other $= 3$, multiracial $= 31$, not reported $= 27$; 2.4% with a psychiatric diagnosis in last 6 months); the Validation was 255 participants (296 before exclusions, 13.9% excluded; age: $M = 33.2$, $SD = 10.6$; sex: female participant $n = 148$, male participant $n = 106$, other $n = 1$; race: white $n = 192$, Black $n = 21$, Asian $n = 14$, Native American $= 2$, multiracial $= 15$, not reported $= 11$; 6.7% with a psychiatric diagnosis in last 6 months).

The Initial sample was used to test a priori hypotheses, as well as to discover new hypotheses. This sample was a re-collection of participants

from a larger study (with different tasks[13]), and so data collection was left open from April 2021 to January 2022 to maximize the overlap between studies. The Validation sample was a new sample, collected to directly replicate effects found in the Initial sample and to test new hypotheses. This work was not preregistered. This sample was collected over a 4-day period in early February 2022. Given the overlap with the height of the COVID-19 pandemic, we ran additional analyses to test for the effects of the pandemic (See Supplementary Note 9 for details).

A power analysis was performed using the Initial sample's results to determine the Validation sample size. To achieve 80% power in detecting a correlation between social distance and the Social Avoidance factor of $r = 0.17$, we estimated a sample size of 222 participants.

### Social navigation task
**Task description**. The social navigation task is a narrative-based social interaction game, where participants interact with different characters in naturalistic settings. Pre-task instructions were minimal: participants were instructed to not overthink their responses and just behave as they would naturally. At the start of the narrative, the participant is told they have just moved to a new town and need to find a job and a place to live. They interact with six illustrated characters over the course of the narrative and occasionally have decision trials in those interactions where their choices could alter the relationships. Each decision trial had two options, which were fully randomized; participants pressed "1" or "2" on their keyboard to select between them. Unbeknownst to the participants, the slides were the same no matter the decisions the participant made; the post-decision slides were written to have narrative continuity regardless of the specific decisions. This ensured that all participants were exposed to the same text. We balanced the gender presentations and skin colors of the characters: in any version, half of the characters were presented as men and half presented as women, as well as half of the characters being darker and half being lighter skinned, which were additionally split between the masculine and feminine characters. We created several balanced versions where the specific characters in the narrative had different demographic characteristics, and randomized participants into a task version. The task was self-paced.

**Affiliation and power decision trials**. The decision trials were categorized as either affiliation or power. Each decision is a choice between two options that (implicitly) move the character in either the negative (−1) or positive (+1) direction along one of the dimensions. There are five characters with six affiliations and six power decisions each, for a total of 60 decisions. Affiliation decisions included whether to share physical touch, physical space, or information (e.g., to share their thoughts on a topic). Power decisions were whether to submit to or issue a directive/command or otherwise exert or give control. See Table 1 for examples of each kind of decision.

**Table 1 | Examples of affiliation and power interaction decisions**

| Decision dimension | Previous slide | Increase decision | Decrease decision |
|---|---|---|---|
| Affiliation | Chris goes in for a hug. | You hug him for a long moment. | You shake his hand instead. |
| Affiliation | Anthony is taking the elevator downstairs with you. | You: "Looks like you're the real person running this place - everybody really relies on you!" | You check your email on your phone: "Is it always that hectic in here?" |
| Affiliation | Mrs. Newcomb gets a phone call. Something is wrong; she seems very worried. | You feel concerned: "Everything alright, Jane? What is it?" | You don't want to get involved: "I'll give you privacy. I think we're done, anyway." |
| Power | Maya: "It doesn't matter if you're late. We're grabbing coffee." | You: "I can't - gotta go. Maybe tomorrow." | You really don't want to be late but reluctantly agree to grab coffee. |
| Power | Kayce is approaching the car. Chris gestures for you to move over to the middle seat. | You wait for Chris to take the middle seat. | You immediately move over to the middle seat. |
| Power | Anthony: "By the way, I have a lead on a place. It's a condo with two separate units." | You: "Get me the contract and floor plan and tell them you need 'til tomorrow." | You: "Let me know when you have more information." |

Several examples of affiliation and power interactions are shown. The slide preceding the choice trial is shown to give the context for the interaction, along with the decision that would increase and the decision that would decrease the location along the relevant dimension. Character gender and names were counterbalanced across participants; a random task version was selected here.

The affiliation and power dimensions were designed to be orthogonal: we wrote the narrative so that both options in every decision were perceived as changing the relationship with the character primarily in terms of affiliation or power, with approximately the same magnitude but with opposite signs. There was also a 6th neutral character with three neutral decisions that did not change their social location; these trials were not included in these analyses.

**Character relationships as affiliation and power trajectories.** We modeled these dimensions together as an Euclidean plane (see Fig. 1). Each character started at the origin, with neutral affiliation and power (0,0). With each decision, that character's latent coordinates were implicitly updated in the positive or negative direction along the current dimension (i.e., each decision moved the character ±1 arbitrary unit along the given dimension, based on the participant's choice). For any given decision trial ($t$), the current character's ($c$) affiliation and power coordinates are the cumulative sums of the trial-wise affiliation and power choices:

$$\text{Affiliation}_{c,t} = \sum_{i=1}^{t} \text{affiliation choice}_{c,i}$$

$$\text{Power}_{c,t} = \sum_{i=1}^{t} \text{power choice}_{c,i}$$

where the choice value is $-1$ or $+1$ if the trial is an interaction of that dimension, and 0 if not. Thus, the affiliation and power location on a given trial equals the affiliation distance and power distance from the origin.

We also represented these locations as the distances and angles ($r$, $\theta$) of the locations relative to the theoretical, "first-person" point-of-view of the participant (by subtracting the maximum possible affiliation and neutral power location: $(6, 0)$)[11]:

$$r_{\text{social}} = \sqrt{(\text{affiliation} - 6)^2 + \text{power}^2}$$

$$\theta_{\text{social}} = \arccos\left(\frac{\text{power}}{\sqrt{(\text{affiliation} - 6)^2 + \text{power}^2}}\right)$$

The angle is in the range [0, 180°]. To capture the participants' overall social tendencies in the task, we calculated the means of these variables. We then z-scored the affiliation, power, and distance values and transformed the angles with the cosine function. For more information on the first-person representation see the supplementary information (Supplementary Note 4).

**Post-task measures**
After completing the social navigation task, participants completed several other related tasks. In the Initial sample, participants completed the memory task followed by the self-reported character placement task. In the Validation sample, participants completed the memory task, followed by (in random order) the self-reported character placement task and the character rating task, and with open-ended questions about the characters last. In each task, the character order (e.g., in the memory options) was randomized. After completing these tasks, participants completed a set of questionnaires, which were presented in a counterbalanced order across participants. These are each described in more detail below.

**Character memory questions.** Participants were asked 30 memory questions about the characters. The correct answers were equally divided between the 6 characters and all 6 characters were provided as options on each question. The order of the questions and the options were

randomized. Once randomized, the option order remained fixed for a given participant to reduce the task burden.

**Self-reported character location placement.** After the memory task, participants completed the self-reported character placement task. We instructed participants about the concepts of affiliation and power and showed them an example of affiliation and power space. The six-character avatars were presented on the bottom left of the screen in a randomized order. Participants were instructed to drag and drop the six-character avatars into the space according to the participant's perceptions of their relationships from the task. The characters were to be placed in this space relative to the participant; the participant's theoretical point-of-view was represented by a red dot at the furthest right end of the affiliation axis (max affiliation) and directly in the middle of the power axis (neutral power). To ensure understanding of the placement task, when participants pressed continue, the program randomly selected two pairs of characters and used a pop-up prompt to ask the participants if their locations reflected the participant's perception of the relationships. For example, the program would ask: "[randomly selected character one] looks more affiliated with you than [randomly selected character two]. Is this where you want them?" Participants then had the option to either reposition the characters or proceed to the next portion of the experiment.

**Open-ended character descriptions.** After the other tasks, participants in the Validation sample reported their impressions of the different characters in open-ended free responses. All characters were presented on the screen at once, in a random order, with free response text boxes on the right of their names and images. Our goal was to capture relatively unstructured self-reported impressions: the instructions included no reference to affiliation or power and were kept minimal to capture the participants' genuine impressions. Qualitatively, participant impressions were varied, with even the same character getting very different descriptions across participants. To illustrate the kinds of responses we received, here are some randomly selected examples across characters: "a 'people-pleaser' who works too hard!", "seems bossy and I felt like you shouldn't get too close to [them]", "outspoken, seems to have a lot of drama in [their] life", "very hard-working, underappreciated", "reliable and steady. friendly but also not afraid to be assertive to get things done."

**Self-reported character ratings.** The Validation participants also completed a self-reported character rating task. Participants were shown images of the characters and asked to rate them on two measures that we expected to capture the meaning of the affiliation and power dimensions in the task. The affiliation interactions involved decisions about friendliness, such as whether to share physical space, touch, or information. We hypothesized these decisions would correlate with how much participants enjoyed interacting with the characters, so we had participants rate how much they liked interacting with each character ("likability"), as well as their perceived real-world likability. The power decisions involved attempts at control, such as accepting, refusing, or making demands. We expected these decisions to relate to perceptions about the characters' relative influence on participants' task-related goals. We had the participants rate the characters' impact on their goals in the game, as well as their perceived own real-world impact and correlated their average power location to the difference between these impact ratings ("relative impact"). Participants rated all characters (in randomized order) on one scale and then completed the other scale; the order of the scales was randomized. All ratings were made on a 100-point scale, with the ends labeled "not at all" and "a lot."

**Self-report questionnaires and factor analysis**
**Questionnaire selection.** After the SNT-related tasks, participants completed a variety of questionnaires. The questionnaire order was counterbalanced across participants. We collected a measure of general

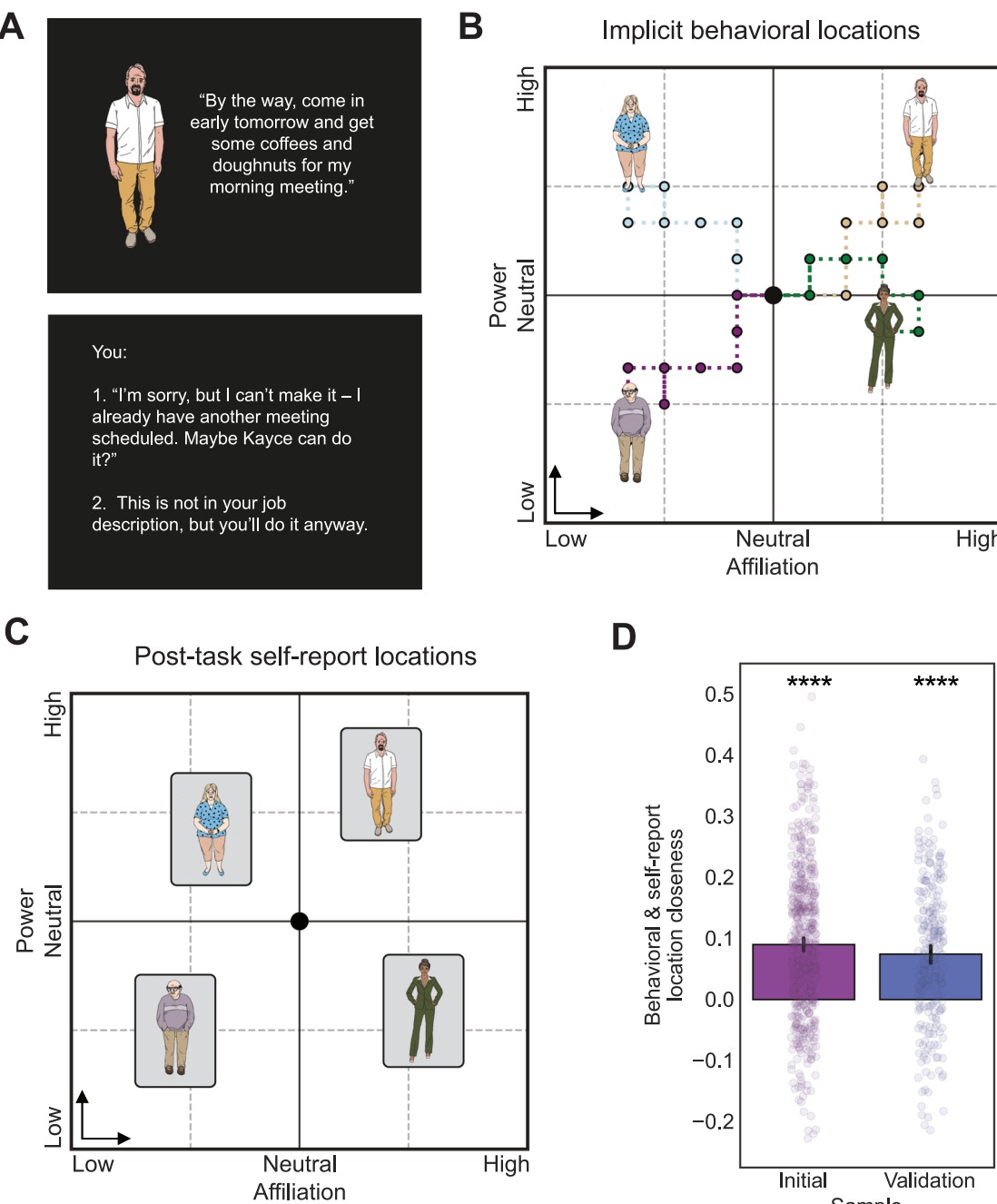

**Fig. 1 | Social interaction decisions form relationship trajectories along abstract dimensions of affiliation and power. A** An example of a power interaction. Participants read the narrative and, on decision trials, choose between two options for how to interact with the current character. The choice implicitly moves the character −1 or +1 along the active dimension (affiliation or power). **B** The participant forms relationships with different characters through sequences of interactions in the narrative. The decisions the participant makes in the implicit affiliation and power interactions change the character's location in social space, forming a relationship trajectory (participants are unaware of the dimensions). Participants interact with 6 characters: 5 each with 6 affiliations, 6 power trials, and 1 with 3 neutral trials. Four characters are shown for illustration. **C** After the task, participants first learned about the affiliation and power dimensions and self-reported the character placements by explicitly placing them into the social space. **D** A permutation analysis shows that the implicit behavioral locations and the explicit self-reported locations are closer than chance, in both samples (Initial sample $n = 579$, Validation sample $n = 255$). The error bars are 2-sided 95% confidence intervals. Right-tailed $p$-value significance is shown with asterisks (**** < 0.001).

cognitive functioning with a truncated version of the International Cognitive Ability Resource (16 items) for an estimate of intelligence quotient (IQ) as the fraction of correct answers. In the Validation sample, we also collected the Social Network Index (SNI), to test hypotheses that we formed in the Initial sample analyses about real-world social network structure. There were several attention-check questions that participants had to correctly answer to be included in the analysis.

We collected questionnaires that we expected to relate to symptoms of social avoidance, mood, and compulsion. We assessed personality disorders with Zanarini Borderline Personality Disorder[14] (referred to as "Borderline Personality") and Avoidant Personality Disorder Impairment Scale[15] ("Avoidant Personality"), social anxiety-related avoidance with the Liebowitz Social Anxiety Scale avoidance subscale[16] ("Social Anxiety"), autism-like deficits with Broad Autism Phenotype Questionnaire[17] ("Autism"),

apathy with Apathy Evaluation Scale ("Apathy"), depression with Zung Self-Rating Depression Scale[18] ("Depression"), obsessive-compulsive disorder with the Obsessive Compulsive Inventory-Revised[19] ("Compulsion") and schizotypy with the Short Scales for Measuring Schizotypy[20] ("Schizotypy"). These 8 questionnaires comprised 177 items.

Our questionnaire approach was inspired by ref. 21, and our questionnaire set overlapped with theirs: we collected the same Social Anxiety, Depression, Apathy, Compulsion, and Schizotypy questionnaires. Previous research shows the fear and avoidance subscales of the Liebowitz Social Anxiety Scale (Social Anxiety) are highly correlated[22]; thus, we only collected the avoidance subscale to reduce participant burden. To ensure broad coverage of social avoidance-related items, we added the Avoidant Personality, Autism, and Borderline Personality questionnaires. Summary statistics for the different questionnaires are presented in the supplementary information (Supplementary Note 1).

We focused on negative constructs, given our interest in transdiagnostic symptoms associated with social avoidance, mood, and compulsivity. However, these response scales capture general tendencies from positive to negative on these constructs, with some items being explicitly framed in positive terms.

In the Initial sample, we collected several other questionnaires that were later discarded; we settled on this specific questionnaire set because analysis in the Initial sample suggested they provide a succinct summary of the participants' task-relevant psychological function.

**Exploratory factor analysis**. We used Exploratory Factor Analysis to find a low-dimensional, latent structure underlying the participants' self-report item responses[21]. This approach addresses challenges with analyzing the effects associated with individual questionnaires. By design, individual questionnaires estimate narrowly defined constructs and are often best suited to specific contexts–a challenge to transdiagnostic research. Analyzing many questionnaires can also cause issues. Multicollinearity between questionnaires means including them in a single model may produce unstable estimates, making it difficult to determine the unique contribution of each questionnaire to the outcome of interest. Further, there is a need for either stringent multiple comparisons correction (where small effects can be missed), or direct replication in an independent sample (where independent noise might produce a new set of unstable estimates and superficially change the pattern of results). Running separate models shifts the multiple comparisons problem to model comparison, and fragments the analysis, making it difficult to draw inferences about the underlying constructs. In contrast, factor analysis finds broad constructs underlying those cross-questionnaire correlations that can offer insight into transdiagnostic symptoms improve replicability, and reduce issues related to multicollinearity and multiple comparisons.

We computed the factors using the Initial sample. We first calculated a pairwise Pearson's correlation matrix between all item-level responses. Latent factors underlying the correlation matrix were estimated with Maximum Likelihood Estimation and then were orthogonally rotated (with Quartimax) to improve factor interpretability.

We retained factors based on Cattell's criterion[23]. This approach sorts eigenvalues in descending order and identifies an "elbow", a sharp drop-off in value where more factors may offer diminishing returns. We implemented this criterion using the Cattell-Nelson-Gorsuch test[24], which calculates the slopes for all possible sets of three adjacent eigenvalues and chooses the factors before where there is the greatest change in slope. This test showed that the difference in slopes is greatest at factor four, suggesting a three-factor solution (see Fig. 2).

To find a smaller set of questionnaire items that capture most of this variance, we selected the items that loaded most strongly and specifically onto each factor and re-ran the factor analysis. We sorted the loadings for each factor and found those items that were above the 25% loading percentile specifically for a single factor. We then re-ran the factor analysis using these items and validated that this reduced set of items produced

similar participant scores as the full set (Pearson's correlation). This approach produced highly similar factor scores as the full factor analysis (all $r$s > 0.92). To test if the factor structure is stable, we re-ran the factor analysis in the Validation sample. The corresponding factors had very high item loading correlations (all $r$s > 0.91), suggesting these three factors capture broad variation across a large set of questionnaire items that replicate across samples. After validating the stability of the factor structure, we transformed the Validation sample's questionnaire responses into the same factors as the Initial sample to ensure the factor loadings were directly comparable across samples.

**Factor labeling**. We labeled each of the factors based on the questionnaires with the highest average loadings (see Table 2). We also calculated word frequencies from the top 20 questionnaire items with the highest loadings, to show the common themes in each factor, after excluding common English stop words (e.g., "the", "and"). To visualize this, we also generated word clouds by scaling the size of each word proportional to its frequency in the top items (see Fig. 2).

We labeled the first factor (i.e., factor with the largest eigenvalue/most explained variance) "Social Avoidance" because the largest loadings came from the Social Anxiety ($M = 0.60$, $SD = 0.10$), Avoidant Personality ($M = 0.55$, $SD = 0.09$) and Autism ($M = 0.53$, $SD = 0.17$) questionnaires. Accordingly, the top questionnaire items were about socializing, especially with new people; for example, the top five items asked the participant if they enjoy "being in social situations," "chatting with people," or if they "look forward to situations where [they] can meet new people", as well as if they avoid "meeting new people" and "going to [parties]". The items were dominated by social words like "people", "talk" and "conversation." Thus, a high score on this factor primarily reflects reports of fearing and avoiding situations with people. This Social Avoidance factor was our factor of interest for the analyses.

We labeled the second factor "Mood" as the highest average loadings came from the Apathy ($M = 0.58$, $SD = 0.13$) and Depression ($M = 0.51$, $SD = 0.17$) questionnaires; the Borderline Personality questionnaire ($M = 0.29$, $SD = 0.08$) was third with a much smaller average loading. The top five Items were about whether the participant has "initiative" and "motivation", and if they "get things done during the day", or "feel that [they are] useful and needed" and "still enjoy doing the things [they] used to do." The most frequent words included "thing" (as opposed to "people" for the Social Avoidance factor), "feel", "important" and "interested".

The third factor was labeled "Compulsion," as it was dominated by the obsessive-compulsive ($M = 0.54$, $SD = 0.04$) questionnaire, followed by borderline personality ($M = 0.30$, $SD = 0.02$) and Schizotypy ($M = 0.29$, $SD = 0.13$) questionnaires. The top items were all obsessive compulsive questionnaire items, e.g.: "'I check things more often than necessary", "I feel compelled to count while I am doing things", "I need things to be arranged in a particular way", "I repeatedly check doors, windows, drawers, etc.", and "I repeatedly check gas and water taps and light switches after turning them off." Like the Mood factor, Compulsion's top words included "thing" and "feel," but also compulsive behavior-related words like "arrange", "repeatedly" and "check".

**Behavioral analyses**

Unless otherwise stated, we ran the analyses in the Initial sample ($n = 579$) prior to collecting the Validation sample ($n = 255$), which was used to directly replicate effects; for these replication analyses, $p$-values were not corrected. Additional hypotheses were formed after analyzing the Initial sample and were only tested in the Validation sample; where appropriate, we correct these $p$-values for multiple comparisons. Given the large sample sizes, we did not formally test linear model assumptions and instead relied on the central limit theorem.

We report statistical results including an estimate of the effect size, 95% confidence intervals, the test statistic, and a $p$-value. For Ordinary Least Squares (OLS) regressions, we report effect sizes as standardized beta values, with confidence intervals computed from the estimated standard errors of

**Fig. 2 | Self-report factor analysis. A** The scree plot shows the twenty largest eigenvalues, with the three-factor solution highlighted in green. These three factors were selected based on when the eigenvalues start to flatten out. **B** Correlation matrix of the self-report questionnaire items (132 items across 8 questionnaires), with the 3-factor structure from the exploratory factor analysis shown on the axes (Initial sample n = 579). The factors are labeled using the questionnaires with the highest average loadings. Questionnaire items are color-coded according to their questionnaire, and questionnaires are labeled according to the symptoms they measure. Avoidant Personality: Avoidant Personality Disorder Impairment Scale; Social Anxiety: Liebowitz Social Anxiety Scale avoidance subscale; Autism: Broad Autism Phenotype Questionnaire; Depression: Zung Self-Rating Depression Scale; Apathy: Apathy Evaluation Scale; Borderline Personality: Zanarini Borderline Personality Disorder; Schizotypy: Short Scales for Measuring Schizotypy; Compulsion: Obsessive Compulsive Inventory-Revised. **C** Word clouds showing the most frequent words from the top twenty most heavily weighted items for each self-report factor, with word size proportional to frequency.

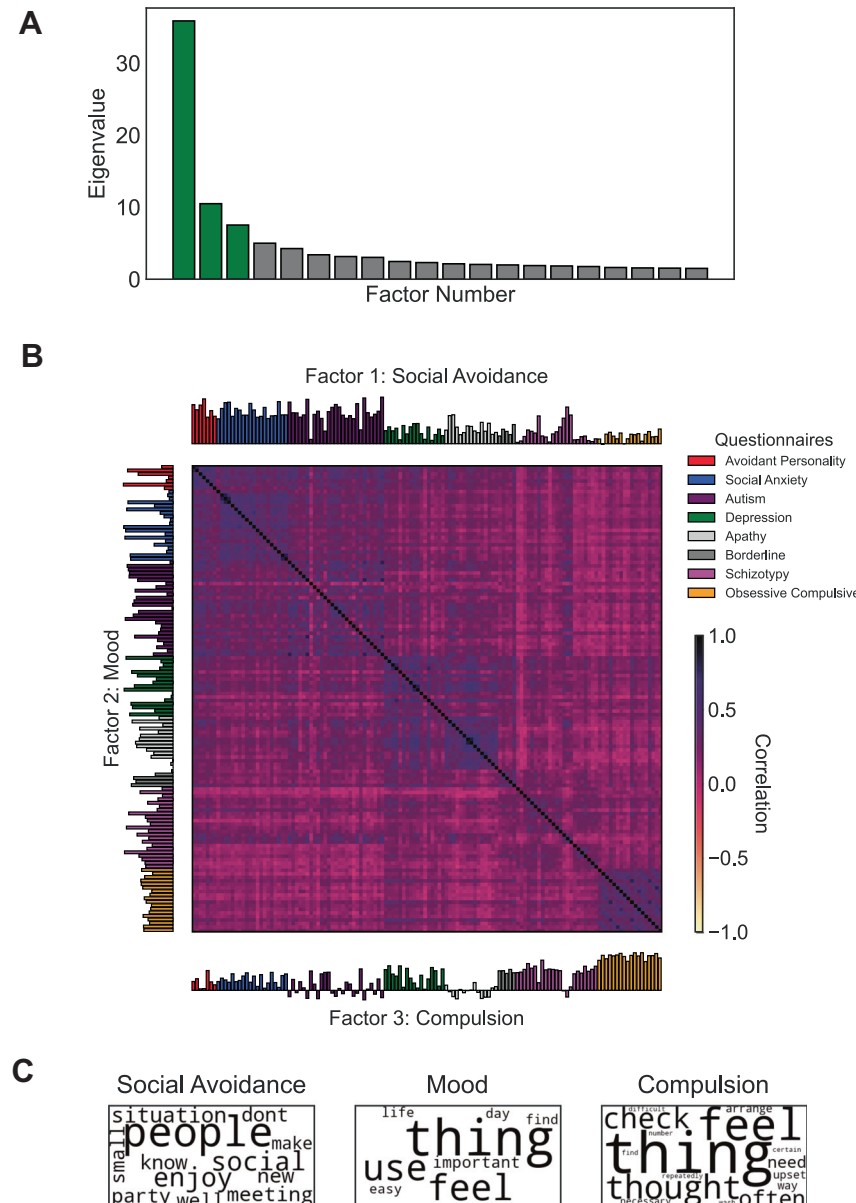

these betas. For one-sample $t$ tests, effect sizes were quantified using Cohen's $d$, and confidence intervals were constructed using the $t$ distribution based on the standard error of the mean. For representational similarity analyses with non-parametric Wilcoxon signed rank tests, we computed the effect size as the mean Kendall's Tau, with confidence intervals derived via 1000 bootstrap iterations.

Control variables included self-reported age (years), sex (male/female), race (white/non-white), diagnosis of a psychiatric disorder within the last 6 months (no/yes), tested IQ (fraction correct) and task memory (fraction correct), and dummy variables for task version. For specific analyses (e.g., sentiment analysis of open-ended character descriptions), additional controls may have been used; these are specified in the relevant methods section.

**Self-reported character location placement analysis.** To validate that the behavioral coordinates reflect the abstract mapping of the social relationships, we tested whether the behavioral and post-task self-reported placement coordinates were closer than expected by chance. We calculated the average character-wise Euclidean distance between the behavioral and self-reported placement locations for a mapping error measure, which we expected the error to be smaller than chance. To

estimate chance, we generated participant-specific null distributions by shuffling the placement locations on the character level and re-calculating the average distance 100 times. We then calculated each participant's mapping error as a standardized score against this null distribution, for a location closeness score:

$$\text{location closeness} = \frac{\mu_{\text{null}} - \text{error}_{\text{map}}}{\sigma_{\text{null}}}$$

This score indicates how many standard deviations the observed mapping was better (positive) or worse (negative) than chance. We tested these scores against 0 using t-tests with right-tailed $p$-values.

We also tested this hypothesis in a complementary way using representational similarity analysis. For each participant, we calculated the pairwise Euclidean distances between the character locations in the task and in the self-report placements and then correlated the non-redundant distances using Kendall's Tau. Across participants, we then tested whether the correlations were larger than 0, using the Wilcoxon signed rank test with right-tailed $p$-values.

**Table 2 | Questionnaire loadings onto the three factors**

| | Social avoidance (factor 1) | Mood (factor 2) | Compulsion (factor 3) |
|---|---|---|---|
| Social anxiety | **0.55 (0.10)** | 0.06 (0.06) | 0.17 (0.08) |
| Avoidant personality | **0.60 (0.10)** | 0.22 (0.06) | 0.13 (0.09) |
| Autism | **0.53 (0.17)** | 0.17 (0.10) | 0.05 (0.15) |
| Apathy | **0.28 (0.11)** | **0.58 (0.13)** | −0.02 (0.08) |
| Depression | 0.22 (0.09) | **0.50 (0.17)** | 0.23 (0.13) |
| Borderline personality | 0.22 (0.07) | **0.29 (0.08)** | **0.30 (0.02)** |
| Obsessive-compulsive | 0.13 (0.07) | 0.12 (0.13) | **0.54 (0.04)** |
| Schizotypy | 0.19 (0.17) | 0.17 (0.09) | **0.28 (0.14)** |

Summary statistics (mean and standard deviations) of the loadings of different questionnaires onto the three factors. Average loadings > 0.25 are bolded to emphasize the dominant questionnaires for each factor.

**Open-ended character description analysis.** To further validate that the participants represent the characters' social locations, we tested whether the semantic representations of the characters in the Validation participants' open-ended text descriptions tracked with the characters' behavioral locations (from participant in-task choices) and self-reported location placements (from participant post-task placements) using representational similarity analysis. For each participant, we generated semantic embeddings for each of the character descriptions using large language models. While these models are the cutting-edge of natural language processing, they can be sensitive to biases in their training data. To ensure our results do not depend on any given semantic space, we used two different models via the Hugging Face interface: all-MiniLM-L12-v2 and all-distilrobert-v1, both finetuned versions of published base models[25,26]. Both models are pre-trained, bi-directional transformer neural networks trained to transform text into contextual embeddings that capture the semantic meaning of sentences.

Each text description was tokenized and passed through the transformer model to generate contextual embeddings for each token. We averaged these embeddings and normalized the result and then calculated the pairwise cosine dissimilarity between the character semantic embeddings. Using Kendall's Tau, we correlated the non-redundant semantic cosine distances with the non-redundant Euclidean distances between the behavioral (or self-reported) placements. We then tested these participant-specific correlation coefficients against 0, using a Wilcoxon signed rank test with right-tailed $p$-values, given that we expected positive relationships.

**Self-reported character rating analysis.** To test hypotheses about the meaning of the behaviors along these dimensions, we correlated (Pearson's r) the average SNT coordinates to the post-task character ratings (in the Validation sample only). We expected dimension-specific correlations: affiliation should correlate more with likability than with relative impact, and power should correlate more with relative impact than with likability. OLS regression with right-tailed $p$-values were used.

**Social avoidance analyses**
**Self-report factor regressions.** After validating social mapping-related assumptions, we tested our main hypotheses about social interaction geometry and self-reported social avoidance using OLS regression. For the main regression analyses, and in each sample separately, the self-report factors were regressed onto the average affiliation and power locations (and control variables):

$$\text{factor score} \sim \text{average affiliation} + \text{average power}$$

We expected the relationships to be negative, so we used left-tailed $p$-values to test the predicted effect. We also tested whether the effects associated with affiliation and power were stronger for the Social Avoidance factor relative to the other two factors, by fitting different OLS models for each combination of individual behavior (affiliation and power) and factor (Social Avoidance, Mood, Compulsion) and comparing the BIC scores. To ensure these effects were stable, we first ran them in the Initial sample and then replicated them directly in the Validation sample.

In the supplementary information, we include additional analyses to contextual our effects. We compared the factors, questionnaire scores, and item responses in their ability to explain task behavior (Supplementary Note 2), we ran additional analyses of the relationships between Social Avoidance and participants' perceptions of their real-world social standing (Supplementary Note 7), and we tested the demographic variables in their ability to explain self-report factors (see Supplementary Note 8).

**Open-ended character sentiment analysis.** To test whether the Social Avoidance factor, social distancing behavior and negative social impressions are related, we conducted sentiment analysis on the Validation participants' free responses about the characters. As with the semantic embedding analysis, we tested this with two separate models to ensure the robustness of the results. We used a widely used rule-based system named VADER[27], which relies on a predefined lexicon and set of rules. We supplemented this model with a large language model fine-tuned to do sentiment analysis[28], which does not depend on the same rule-based assumptions and may be better suited to capture contextual nuances of language.

Prior to analyzing the text responses with VADER, we cleaned the text responses by converting the text to lowercase, removing punctuation, and lemmatizing. For both models, we computed the mean compound sentiment score across characters, which summarizes the overall sentiment of a text as the weighted sum of sentiment scores (negative, neutral, positive), normalized to be between −1 (most negative) and +1 (most positive). We predicted that the average compound sentiment would negatively correlate with both the average social distance and the Social Avoidance factor score. We tested this prediction with OLS regression and a left-tailed $p$-value. We also included the Mood and Compulsion factor scores in the model, the standard controls and average word count as an extra control variable.

**Real-world social network analyses.** After running the factor analysis regressions in the Initial sample, we hypothesized that if the average social distance from the participant to the characters tracks with real-world social distancing behaviors, it should also relate to smaller and less complex real-world social networks. To test this in the Validation sample, we correlated two SNI subscales with social distance: social network size (how many people they interact with at least every 2 weeks) and social network diversity (how many categories of relationships [family, friend, religious, etc.] they have). We ran OLS regressions and corrected the $p$-values for 2 comparisons using Bonferroni's method to control the family-wise error rate (FWER).

**Reporting summary**
Further information on research design is available in the Nature Portfolio Reporting Summary linked to this article.

# Results
## Decision-making shaping social interactions is consistent with self-reported social mapping
The social navigation task aims to capture how people represent and navigate social relationships along two fundamental dimensions: affiliation and power. The task is a narrative-based game where participants make a series of decisions about how to interact with different characters in naturalistic social situations. Unbeknownst to the participants, each choice implicitly moves the character along either an affiliation or power dimension in abstract social space, creating social trajectories (see Fig. 1). We tested whether these

behavioral locations reflect subjective social mapping by analyzing three different post-task measures: self-reported placements of the characters onto an affiliation and power space to validate the location representation, free response semantic representations of the characters for convergent evidence of location representation, and self-reported ratings of character liking and relative impact to validate the social dimensions themselves.

**Behavioral affiliation and power locations relate to self-reported character location placements.** To test if the participants represent the social locations implied by their behavioral choices, we asked whether self-reported (i.e., explicit) placement of characters in social space align with the behavioral (i.e., implicit) locations from task decision (see Fig. 1). After the social navigation task, we had participants place the characters onto an affiliation and power space according to their own impressions of the characters (this was the first time they became aware of these dimensions) and then tested if the behavioral and self-reported locations were correlated. The behavioral and self-reported location placements were on average closer in social space than chance (t-tests; Initial sample: Cohen's $d = -0.78$, CI$_{95\%}$ = [−0.88, −0.71], $t_{578} = -18.2$, $P < 0.001$; Validation sample: Cohen's $d = -0.66$, CI$_{95\%}$ = [−0.78, −0.53], $t_{254} = -10.61$, $P < 0.001$). Representational similarity analysis showed that the patterns of distances between characters were also similar between the behavioral and self-reported locations (Initial sample: mean tau = 0.07, CI$_{95\%}$ = [0.06, 0.09], $W = 97610$, $P < 0.001$; Validation sample: mean tau = 0.08, CI$_{95\%}$ = [0.06, 0.11], $W = 22072$, $P < 0.001$). These results suggest participants represent these relationships along the affiliation and power axes, despite the dimensions never being explicitly mentioned in the pre-task instructions or narrative.

**Behavioral and self-reported affiliation and power locations relate to verbal descriptions of the characters.** To offer converging evidence for social location representations, we tested whether the Validation participants' open-ended descriptions of the characters reflect their affiliation and power locations. Using large language models, we embedded each participant's character-specific text-free responses as semantic vectors and then correlated the pairwise semantic distances with their pairwise behavioral location distances and pairwise self-reported placement distances. In other words, we asked whether characters with similar semantic representations also have close locations in the two-dimensional space. In two different language models, the semantic distances were correlated with both the behavioral (mean tau = 0.04, CI$_{95\%}$ = [0.01, 0.07], $W = 16664$, right-tailed $P = 0.011$; mean tau = 0.02, CI$_{95\%}$ = [0.00, 0.06], $W = 15573$, right-tailed $P = 0.03$) and the self-reported placement distances (mean tau = 0.11, CI$_{95\%}$ = [0.07, 0.13], $W = 23044$, right-tailed $P < 0.001$; mean tau = 0.07, CI$_{95\%}$ = [0.02, 0.08], $W = 19540$, right-tailed $P = 0.003$). This analysis offers converging evidence that participants' self-reported representations contain information about affiliation and power.

**Behavioral affiliation and power dimensions relate to self-reported ratings of character liking and impact.** Having established that participants represent the characters' social locations both explicitly (through social space placements) and implicitly (through open-ended text descriptions), we next sought to establish whether these dimensions map onto theoretically relevant social judgments. We predicted that affiliation choices should correlate with liking the characters, while power choices should correlate with the perceived relative impact of the characters on one's own goals. To test this, we had Validation participants rate how much they liked interacting with each character ("likability"), and how much control or influence they perceived each character to have relative to themselves ("relative impact"). Using regressions, we found a double dissociation: average character likability correlated with the average affiliation location ($\beta = 0.40$, CI$_{95\%}$ = [0.29, 0.52], $t_{232} = 6.92$, right-tailed $P < 0.001$)–but

not power ($\beta = 0.04$, CI$_{95\%}$ = [−0.08, 0.17], $t_{232} = 0.65$, $P = 0.51$)–and average relative impact correlated with the average power location ($\beta = 0.20$, CI$_{95\%}$ = [0.08, 0.33], $t_{232} = 3.18$, right-tailed $P < 0.001$)–but not affiliation ($\beta = -0.01$, CI$_{95\%}$ = [−0.14, 0.12], $t_{232} = -0.22$, $P = 0.82$). Analyses of other social judgments confirmed the relative specificity of these relationships (see Supplementary Note 5). Thus, the social interaction tendencies of the participants related to conceptually meaningful dimensions: participants who liked the characters made more affiliative decisions, and participants who rated themselves as having a higher impact (i.e., influence on goals) than the characters made more dominant decisions.

Together, these analyses provide converging evidence that the social navigation task captures how people represent and navigate relationships along the axes of affiliation and power. Participants' affiliation and power choices were evident in their explicit spatial mappings, open-ended text descriptions, and social judgments of the characters—despite the underlying affiliation and power dimensions never being explicitly mentioned in the task.

## Social Avoidance has a two-dimensional behavioral structure: low affiliation and low power

**Low affiliation and low power behaviors independently relate to the Social Avoidance factor.** We then turned to social navigation task behavior to test our main hypotheses. We first tested whether the Social Avoidance factor score negatively correlated with affiliation and power decisions in the simulated social interactions (**hypothesis 1**). We used OLS regression: for each factor, the scores were regressed onto the average affiliation power locations, and control variables. As predicted, we found that the Social Avoidance factor negatively correlated with average affiliation location and with average power location, in both the Initial sample (affiliation: $\beta = -0.22$, CI$_{95\%}$ = [−0.31, −0.14], $t_{555} = -5.25$, left-tailed $P < 0.001$; power: $\beta = -0.14$, CI$_{95\%}$ = [−0.22, −0.06], $t_{555} = -3.26$, left-tailed $P < 0.001$) and Validation sample (affiliation: $\beta = -0.25$, CI$_{95\%}$ = [−0.37, −0.12], $t_{231} = -3.78$, left-tailed $P < 0.001$; power: $\beta = -0.12$, CI$_{95\%}$ = [−0.25, 0.01], $t_{231} = -1.377$, left-tailed $P = 0.039$). The effects were highly similar whether affiliation and power were included in the same model or separate models, suggesting that the two dimensions have independent relationships with the Social Avoidance factor. Moreover, control analyses show that each character showed the same pattern of affiliation and power effects, suggesting the avoidance effect was consistent across relationships–regardless of the character's specific temporal position or social role in the narrative (see Supplementary Note 3). Thus, our main predictions were supported: social avoidance-like symptoms relate to the specific social interaction styles of low affiliation and low power (see Fig. 3).

**Social Avoidance factor specifically has a strong relationship with affiliation and power.** To test for the specificity of these effects, we tested whether the Social Avoidance factor showed stronger effects with affiliation and power compared to the Mood and Compulsion factors. We ran OLS regressions for each combination of behavior (average affiliation and power) and factor (Social Avoidance, Mood, Compulsion), and compared the BIC scores, where positive BIC differences ($\Delta$BIC) indicate stronger evidence for Social Avoidance relative to the comparison factor. Across both samples, all $\Delta$BIC values favored Social Avoidance when compared to both Mood (Initial sample: affiliation $\Delta$BIC = 17.88, power $\Delta$BIC = 6.44; Validation sample: affiliation $\Delta$BIC = 9.74, power $\Delta$BIC = 2.51) and Compulsion (Initial sample: affiliation $\Delta$BIC = 20.4, power $\Delta$BIC = 6.34; Validation sample: affiliation $\Delta$BIC = 9.88, power $\Delta$BIC = 2.52) (see Fig. 3). Heuristically, these BIC differences are meaningful: a difference of 2 or more offers positive evidence in favor of the model with smaller BIC, with larger differences providing stronger evidence[29]. As such, our hypothesis of specific relationships between affiliation and power and Social Avoidance was supported (**hypothesis 2**).

**Fig. 3 | Effects of self-report factors and task-based social relationship geometry. A** As predicted, low affiliation (blue; less affiliation between participant and characters) and low power (red; more power to the characters relative to self) behaviors are related to high social avoidance-like symptoms. Affiliation and power behaviors were included in each model, with controls (age, sex, race, IQ, presence of psychiatric diagnosis, task version, and task memory), predicting the different self-report factors. The bar heights are Ordinary Least Squares beta values (e.g., the effect related to average affiliation, controlling for average power and the control variables), the error bars are 2-sided 95% confidence intervals, and the asterisks are p-value significance (* < 0.05, ** < 0.01, *** < 0.005, **** < 0.001). Left-tailed p-values were used for the social avoidance factor as our hypotheses were directional. These effects were similar in both samples (Initial sample n = 579, validation sample n = 255), and whether affiliation and power were modeled together or separately. **B** Social Avoidance had the smallest Bayesian Information Criterion (BIC) score for predicting the affiliation and power tendencies in both samples, suggesting the relationship between task behavior and self-reported factors was relatively specific to the Social Avoidance factor.

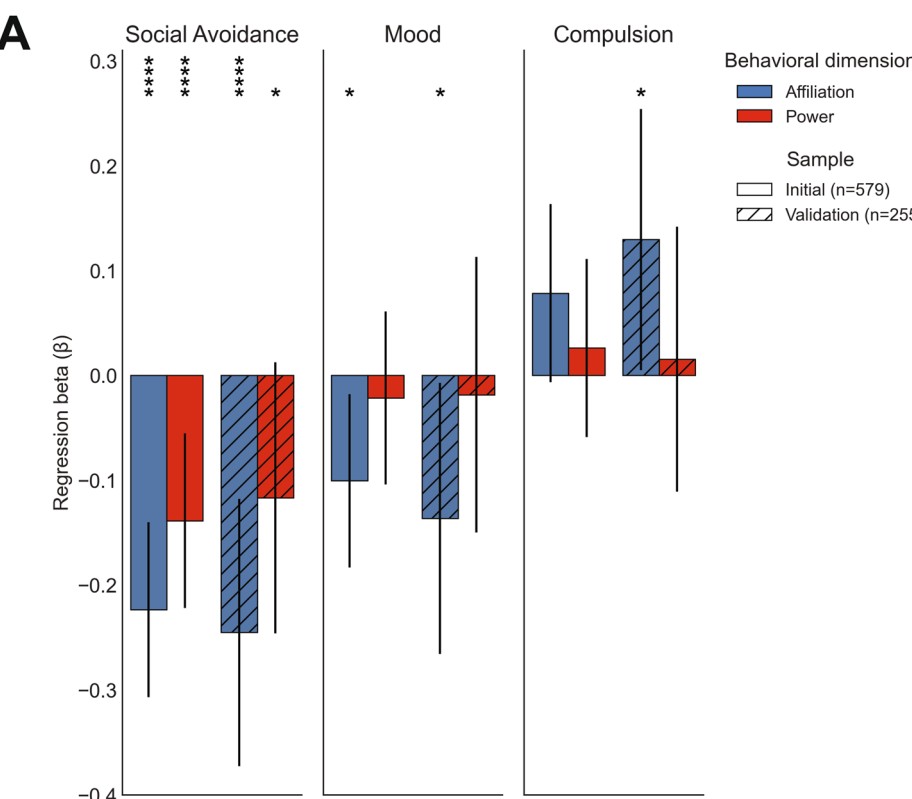

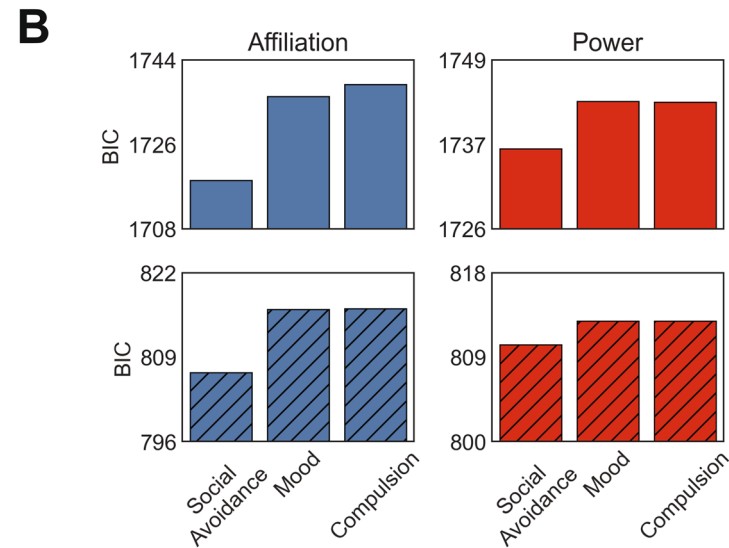

## Social Avoidance factors and behavioral social distance relate to emotion about the characters

We hypothesized that individuals with higher Social Avoidance scores would form more negative impressions of the characters, and that this would be reflected in both their behavior and text descriptions of the characters. Having already established that both affiliation and power independently relate to social avoidance, we used a social distance metric that combines the participant's average affiliation and power decisions, calculated as the length of the vector between the participant's point-of-view and character locations in the two-dimensional social space (see Supplementary Notes 4 and 6). We tested whether participants with higher self-reported Social Avoidance and larger behavioral social distance reported more negative impressions of the characters in their open-ended text descriptions (**hypothesis 3**).

We analyzed the sentiment expressed in the Validation sample's open-ended character descriptions using two different sentiment analysis models–a rules-based model and a large language model. Average sentiment related to social distance in both models, with larger distance relating to more negative emotion (rules-based model: $\beta = -0.25$, $CI_{95\%} = [-0.37, -0.12]$, $t_{231} = -3.8$, left-tailed $P < 0.001$; language model: $\beta = -0.29$, $CI_{95\%} = [-0.41, -0.16]$, $t_{231} = -4.5$, left-tailed $P < 0.001$). In a regression with all three self-report factors, average sentiment also specifically and negatively correlated with the Social Avoidance factor (rules-based model: Social Avoidance: $\beta = -0.18$, $CI_{95\%} = [-0.31, -0.06]$, $t_{229} = -2.85$, left-tailed $P = 0.0047$, Mood: $\beta = -0.11$, $CI_{95\%} = [-0.24, 0.03]$, $t_{229} = -1.59$, $P = 0.11$, Compulsion: $\beta = -0.01$, $CI_{95\%} = [-0.15, 0.12]$, $t_{229} = -0.19$, $P = 0.84$; language model: Social Avoidance: $\beta = -0.19$, $CI_{95\%} = [-0.32, -0.06]$, $t_{229} = -2.97$, left-tailed $P = 0.0017$, Mood: $\beta = -0.14$, $CI_{95\%} = [-0.27, 0.01]$, $t_{229} = -2.1$, $P = 0.0361$,

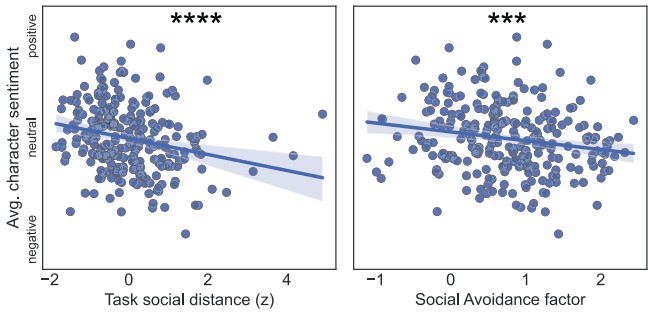

**Fig. 4 | Average free response sentiment about the characters correlated with both task-based social distance and social avoidance factor.** The sentiments expressed in post-task open-ended responses correlated with both the average task-based social distance and the score on the social avoidance factor (tested in the validation sample; $n = 255$). Higher compound sentiment indicates more positive sentiment; as compound sentiment increases, social distancing behavior decreases. Simple OLS regressions are shown, with 95% confidence intervals; FWER-corrected left-tailed $p$-values from the full models (including covariates) are indicated as asterisks (*** < 0.005, **** < 0.001).

Compulsion: $\beta = 0.04$, $CI_{95\%} = [-0.09, 0.17]$, $t_{229} = 0.55$, $P = 0.58$). These free-response sentiment results support our hypothesis: negative perceptions of others are related to social avoidance and its behavioral manifestation in relationships (see Fig. 4).

### Social Avoidance factors and behavioral social distance relate to real-world social network structure

If behavioral tendencies in the social navigation task reflect real-world social tendencies, task social distance should relate to real-world social distance, which may, in turn, relate to smaller real-world social networks. This may be true regardless of whether the social distance is generally high power or low power: a tendency towards greater social distance creates fewer social relationships. Thus, after analyzing the Initial sample, we hypothesized that task social distance should correlate with the structure of real-world social networks (**hypothesis 4**).

To test this, we collected the SNI[30] in the Validation sample and ran regressions to test relationships with two measures of network structure: the number of people regularly interacted with (network size) and the number of different kinds of relationships (network diversity). First, we validated that the Social Avoidance factor relates to real-world social networks: as expected, both social network size and diversity had strong negative correlations with the Social Avoidance factor (separate OLS models; size: $\beta = -0.34$, $CI_{95\%} = [-0.46, -0.22]$, $t_{232} = -5.58$, left-tailed $P_{FWER} < 0.001$; diversity: $\beta = -0.3$, $CI_{95\%} = [-0.42, -0.18]$, $t_{232} = -4.92$, left-tailed $P_{FWER} < 0.001$). Also, as predicted, task social distance negatively correlated with both real-world social network measures (separate OLS models; size: $\beta = -0.14$, $CI_{95\%} = [-0.27, -0.02]$, $t_{232} = -2.23$, left-tailed $P_{FWER} = 0.0135$; diversity: $\beta = -0.14$, $CI_{95\%} = [-0.27, -0.01]$, $t_{243} = -2.14$, left-tailed $P_{FWER} = 0.0166$), supporting our hypothesis that individuals who socially distance themselves in the task have smaller and less diverse networks (see Fig. 5). Thus, both self-reported Social Avoidance and abstract social distancing behavior are related to real-world social network structure.

### Discussion

In this study, we report the results from two large-scale online samples on the structure of social avoidance behavior in naturalistic social interactions and real-world social networks. We demonstrate that a Social Avoidance factor in the general population has a two-dimensional structure of affiliation and power—two fundamental social dimensions. Moreover, we show these relationships in actual social behavior rather than solely in self-report and introduce a behavioral paradigm for studying naturalistic social behavior.

Using a self-report questionnaire factor analysis and geometric measures of simulated social relationships, we found that high self-reported social avoidance relates to low affiliation and low power social interaction decisions, independently (**hypothesis 1**). This effect was specific to the Social Avoidance factor (**hypothesis 2**). This style of interaction increases the abstract social distance between the participant and the characters, by reducing their affiliation and relative power with others. We found further evidence for this interpretation in participants' open-ended text responses after the task: participants higher in the Social Avoidance factor (and who created more abstract social distance in the task) were more likely to express negative sentiment about the characters (**hypothesis 3**).

Moreover, task-based abstract social distance correlates with *real-world* social network structure: more social distance relates to a smaller and less diverse social network in the real world (**hypothesis 4**). These effects replicated and were remarkably similar in independent samples (Initial sample $n = 579$, Validation sample $n = 255$). Thus, self-reported social avoidance symptoms may reflect multidimensional behavioral tendencies—a navigational tendency in abstract social space.

### Social avoidance as multidimensional navigation

The multidimensionality of Social Avoidance behaviors fits with previous work. Theoretical accounts[31] and empirical studies[8] have suggested the importance of both affiliation and power in social avoidance, but, to our knowledge, ours is the first study on social avoidance to study these dimensions together as behaviors. Previous studies used self-reports[32] or responses to simple stimuli, such as faces[33]; while these approaches can provide useful information, these studies had mixed results, perhaps because they did not measure naturalistic behavior. For example, while some work suggested social avoidance is related to both self-reported affiliation and power[32], other work suggested a relationship with power but not affiliation[9]. Here, we leveraged theoretically motivated hypotheses with big data, data-driven quantification of symptoms, and geometric measures of naturalistic behavior. In doing so, we showed a clear two-dimensional affiliation and power behavioral structure to social avoidance, with additional relationships to semantic representations in language and the structure of real-world social networks.

Could we find neural distortions corresponding to these behavioral differences? Previous work with this social navigation task suggests we can[11]: functional magnetic resonance imaging (fMRI) activity in the hippocampus that tracked relationships along affiliation and power also correlated with self-reported social avoidance, suggesting the neural mechanisms of social mapping may be implicated in social avoidance. In this study, we treated the dimensions as orthogonal and fixed; but with neural data, we could ask questions about the representations of the shape of social space itself. It may be that people high in social avoidance have a restricted social map, with a relatively lower number of possible locations and trajectories. Predictions of negative outcomes are also central to social avoidance: do people high in social avoidance simulate two-dimensional relationship trajectories related to negative social outcomes? Again, the hippocampus may play a key role in tracking these trajectories[1,12,34,35].

### Limitations

One strength of our approach is that we have two large samples that represent a broad swath of the general population. However, as we did not have any clinical assessments, and we had a low prevalence of individuals with psychiatric diagnoses in the last 6 months, we were limited to learning from participants' responses to self-report questionnaires. Future work should recruit clinical samples (e.g., with a DSM diagnosis of Social Anxiety Disorder) to test for the same relationships we found here. Moreover, multiple disorders show social avoidance-like behaviors, and there are other social symptoms (e.g., communication deficits) that could be explored using this task. Future work should also consider alternative constructs, beyond self-reported social avoidance (e.g., sensitivity to social norms and social context).

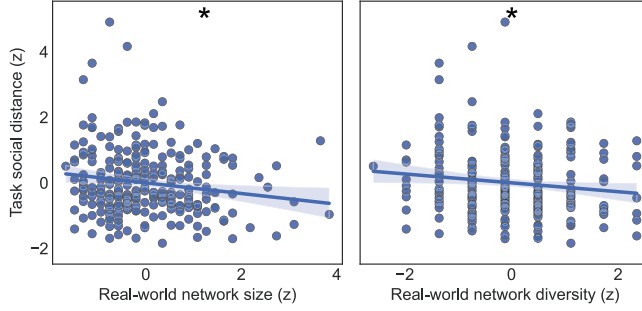

**Fig. 5 | Real-world social network structure correlates with task-based social distance.** The first-person social distance variable negatively correlates with measures of real-world social network size and relationship diversity (tested in the validation sample; $n$ = 255). Simple OLS regressions are shown, with 95% confidence intervals; FWER-corrected $p$-values from the full models are indicated as asterisks (* < 0.05). Accordingly, participants who had larger distances from task characters also tended to have real-world networks smaller in size and diversity.

## Future work: clinical samples and social narratives

Another strength of our approach is the use of a naturalistic and easy-to-understand task to study first-person social interaction. By tracking the individual's social decision-making as a navigational vector in affiliation and power space, we can simplify complex questions about social interactions and relationships down to abstract navigation. The power of this approach, in combination with data-driven analysis of large-scale self-reports, is shown here. The approach could be further developed to test new ideas. For example, does increasing the ambiguity of the characters' behaviors or the task goals increase the social distancing behaviors of socially avoidant individuals?

Could pairing more realistic, and perhaps even personalized, social interactions (e.g., with chatbots) along with feedback help individuals learn more adaptive social tendencies? Deploying multiple narratives, or a single extended narrative, over time could be used to estimate treatment effects. Many such questions are possible.

In conclusion, the altered behavioral geometry of individuals with high Social Avoidance factor scores suggests that social avoidance and social interaction, more generally, can be thought of as an abstract social navigation strategy.

## Data availability

All data are available at https://osf.io/yvts6/.

## Code availability

Data were analyzed with Python 3 scripts. The code to reproduce the analyses is available at https://github.com/matty-gee/social_avoidance.

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

## Acknowledgements

M.S. was supported by the National Institute of Mental Health (grant numbers T32MH015144, F31MH123123). D.S. was supported by the National Institute of Mental Health (grant numbers R21MH120789, R01MH122611, R01MH123069) and SFARI Award 877761. The funders had no role in study design, data collection and analysis, the decision to publish, or the preparation of the manuscript. This work was supported in part through the computational and data resources and staff expertise provided by Scientific Computing and Data at the Icahn School of Medicine at Mount Sinai and supported by the Clinical and Translational Science Awards (CTSA) grant UL1TR004419 from the National Center for Advancing Translational Sciences. Research reported in this publication was also supported by the Office of Research Infrastructure of the National Institutes of Health under award numbers S10OD026880 and S10OD030463. The content is solely the responsibility of the authors and does not necessarily represent the official views of the National Institutes of Health.

## Author contributions

M.S. and D.S. conceptualized the study. M.S. collected the data and ran the analyses. M.S. wrote the original draft, which D.S. reviewed and edited.

## Competing interests

The authors declare no competing interests.
