## [Transparent Peer Review file · Communications Psychology]

Social avoidance can be quantified as navigation in abstract social space

Corresponding Author: Dr Matthew Schafer

Version 0:

Decision Letter: first round

Dear Dr Schafer,

Thank you for your patience during the peer-review process. Your manuscript titled "Social Avoidance as Multidimensional Navigation" has now been seen by 2 reviewers, whose comments are appended below. You will see that they find your work of some potential interest. However, they have raised quite substantial concerns that must be addressed. In light of these comments, we cannot accept the manuscript for publication, but would be interested in considering a revised version that fully addresses these serious concerns.

We hope you will find the Reviewers' comments useful as you decide how to proceed. Should additional work allow you to address these criticisms, we would be happy to look at a substantially revised manuscript. If you choose to take up this option, please highlight all changes in the manuscript text file, and provide a detailed point-by-point reply to the reviewers.

Editorially, we consider it important that you can demonstrate the specificity of the social navigation measures and that they are not correlated with other dimensions related to social interactions. We also consider it important to provide further methodological details to address the reviewers concerns regarding alternative explanations for your findings.

I am attaching a checklist that details critical reporting requirements for the revised manuscript. Please attend to each item and ensure your manuscript is fully compliant. We are requesting that your manuscript aligns with these requirements as this facilitates the evaluation of your manuscript, reducing delays in re-review and potential future acceptance. If your revised manuscript is not aligned with these requests on major issues, such as those concerning statistics, it may be returned to you for further revisions without re-review. Additional information can be found in our style and formatting guide Communications Psychology formatting guide.

If the revision process takes significantly longer than five months, we will be happy to reconsider your paper at a later date, provided it still presents a significant contribution to the literature at that stage.

Please use the following link to submit your

- revised manuscript,
- point-by-point response to the referees' comments,
- cover letter (as a separate document),
- the Editorial Policy Checklist (see below),
- the Reporting Summary (see below), and

- the completed Editorial Request Table (attached):

Link Redacted

Thank you for the opportunity to review your work.

Best regards,

Patricia Lockwood

Patricia Lockwood, PhD
Editorial Board Member
Communications Psychology
orcid.org/0000-0001-7195-9559

REVIEWER EXPERTISE:

Reviewer #1 Social cognition, decision-making, navigation and cognitive maps

Reviewer #2 Social decision-making

REVIEWER REPORTS:

Reviewer #1 (Remarks to the Author):

This study aims to test whether the trait of social avoidance specifically influences how individuals are placed in a conceptual social space based on their experiences of social interactions. This is an intriguing approach, especially considering the potential for leveraging interactions between neural representations of social relationships and everyday social behaviors to develop behavioral training for certain clinical populations. The manuscript presents a comprehensive study utilizing advanced natural language processing techniques and factor analysis. However, many arguments are difficult to falsify, and several critical areas require clarification and improvement.

1. This study utilizes many cutting-edge natural language processing (NLP) techniques. I believe it is important to report the potential limitations of these methods and consider supplementing them with additional methods or tools.
2. "Based on the introduction, I anticipated that the authors would analyze the impact of the social avoidance trait on trial-by-trial decisions during social interactions. However, the study design appears to measure decisions associated with low-affiliation and low-power tendencies after the social interactions, only once. This discrepancy between the introduction's expectations and the actual study design raises concerns about the validity of the hypothesis, findings, and interpretations. For example, the statement 'people highest in these symptoms should make low-affiliation and low-power decisions in the task's social interactions'; Similarly, the statement 'Here we sought to test the extent to which people high in self reported social avoidance show implicit low-affiliation and low-power behavioral tendencies in social interactions' needs to be re-evaluated considering the timing of the decision-making process.
3. Could a single character's low affiliation or power reduce the average affiliation and power tendencies across all characters? It's unclear whether this would result in a general shift in the 2D social space, or if only certain characters within specific domains are more likely to be pulled towards lower values.
4. What is the range and variance of the responses for each clinical questionnaire? Can the authors compare these values with those from diagnosed populations? It is unclear to what extent the current participant sample is representative and whether the current data can be generalized to potential clinical populations.
5. The manuscript's explanation of the differences in explainability between the polar coordinate model (first-person point-of-view) and the Cartesian model is unclear. While the angle and Euclidean distance in the polar coordinate system theoretically contain the same information as the position on the affiliation and power axes in the Cartesian system (assuming a shared origin), the manuscript does not adequately explain how the polar coordinate values are computed or why this leads to different levels of explainability. To clarify the differences in explainability, the manuscript should provide a detailed explanation of how the angle and Euclidean distance are calculated in the polar coordinate model. This should include the specific formulas used and how they relate to the affiliation and power axes. Moreover, this will be appreciated if the authors can clearly articulate the specific reasons why the polar coordinate model might offer a different level of explainability compared to the Cartesian model.
6. The BIC values for the different models appear to be quite similar. Was the overall BIC calculated by summing the individual BIC values for each participant? If so, how many participants exhibited a lower BIC for the target model compared to the other models?
7. Please provide a Scree plot in the factor analysis and provide a more specific rationale behind the choice of three factors.
8. The readers would appreciate it if the authors could specify the questions corresponding to the X-axis, as well as the scales and scores on the Y-axis in Figure 3. Currently, it is unclear and less intuitive why the authors have arranged the factor scores in this particular manner within the cross-correlation graph. What information can be derived from the cross-correlation graph in Figure 3? Can the authors provide more details on the insights that readers should gain from this figure?

Reviewer #2 (Remarks to the Author):

The current study aims to provide a framework for the investigation of social avoidance that extends from mere self-reports. Authors claim that a navigational vignette task can capture how people's self-reported social avoidance can be directly connected to low affiliation and low social power. Secondly, they claim that this effect is selectively a function of their social avoidance level as measured by a series of self-report questions. Finally, they claim to capture not only social avoidance as a task-related construct, but that this can be extrapolated to real life behaviour, and thus could have predictive validity.

I find the approach of the paper both innovative and relevant. The suggested behavioral paradigm offers a novel exploration of the investigated concept that can be improved and adjusted to match also the exploration of other concepts that may follow a similar pattern of association. While the experimental task does use vignettes and scenarios, the idea of designing a cartesian system in order to track navigational patterns I find interesting and potentially influential. Provided that the underlying methodology supports the findings, I believe the idea of connecting real social avoidance to a spatial-navigational component is definitely interesting.

However, I am not convinced that the vignettes and scenarios are able to achieve this goal. While it is a non-trivial problem to move from highly simplified and controlled tasks to more ecologically valid tasks, I worry about potential alternative explanations of what the Social Navigation Task might capture.

Below, several points of concern:

1. In the questionnaires used there are factors addressing challenging aspects related to social interactions, however, I think it would have been useful to also have some positive valenced self-reports in order to verify that there is no underlying aspect of social orientation, social skill, or social norm sensitivity that might affect the interpretation of the results.
2. While I find the task and 2D mapping innovative, I am missing the crucial link. From my understanding of the paper, I am not convinced that the task really isolates features of social avoidance, but rather gets at the likeability of characters, beliefs about social norms and appropriate, or acceptable social behaviour in such contexts. I would like to see if, as mentioned above, the social navigation measures are not in fact correlated with other dimensions related to social interactions. Also, it is not clear for me from the presentation of the results, how connected are the questionnaires of social avoidance to the score in the task.
3. All secondary analyses related to different aspects of the question at hand, could be labeled as "exploratory" and be included in the supplementary material. This would make the results less confusing and improve the flow of the manuscript.
4. If more standardized data visualizations related to each type of analyses and results were included, that could help. Most of the ones that are included seem to me non-intuitive in supporting findings. Too many different facets of findings are presented based on different types of analyses and plots. More consistency and clarity regarding the type of analysis and the chosen visualization for each hypothesis could help.
5. It would improve the manuscript if a more solid justification (based on existing literature) regarding the rationale behind the choice of each of the batteries used to measure specific traits of the participants, and why this is limited to "negative" constructs. Explaining, also, why the choice to use specific dimensions and subscales of each battery for their hypotheses (maybe a pre-registration could have strengthened these choices). A better definition of the concepts referred to. The switch between whether the measured values were impressions, emotions or sentiments, as the authors do not clarify this sufficiently prior to presenting the results. More consistency in terminology could help the reader.
6. Currently, the theoretical background of the question at hand reads a bit weak and confusing. For example, the two main concepts (i.e., affiliation and power) are not clearly defined in the paper. The explanations provided "Affiliation decisions included whether to share physical touch, physical space, or information (e.g., to share their thoughts on a topic). Power decisions were whether to submit to or issue a directive/command, or otherwise exert or give control." seem insufficient to me. Furthermore, based on the example given, it is unclear what "sharing physical touch or information" means (see also the comment below regarding the example included). This may generate confusion as (social) power definition and measurement, for instance, varies based on the scientific discipline (i.e., social psychology: influence the behavior, emotions, or beliefs of others; organizational psychology: capacity of leaders to influence the behavior of their followers to achieve organizational goals; experimental economics: control over the resources of another). The same applies for affiliation.

I believe that the amount of details in the methodology description could be sufficient for a replication and the statistical analysis is in order. However, it would be helpful to get a clearer justification why the specific analysis methods were chosen, and what each could add. Ideally, I would limit the included analyses per hypothesis presenting only statistical methods and associated results that answer the main question(s). The presented analysis seems to be more exploratory, or at least not immediately embedded in practices of the field. Moreover, it is not clearly connecting the methodology used to the RQs and the hypotheses. A more detailed explanation, and backing up of analysis methodology and results could help improve the paper.

Overall, both the underlying idea and the novel experimental paradigm used are promising. The authors should be encouraged to improve their theory, relate it substantially with the existing literature on the field and provide conceptual clarity regarding their main concepts and measurements. In addition, it is strongly recommended that, in the future, they pre-register subsequent studies on the topic, so that their hypotheses are based on confirmatory analyses.

A resubmission should contain the clarification of the connection between the social avoidance and the task is clarified, together with the motivation regarding the analysis methods for the main research question.

EDITORIAL POLICIES

We ask that you ensure your manuscript complies with our editorial policies and reporting requirements.

To that end, we require revised manuscripts to be accompanied by two completed items: a reporting summary that collects information on study design and procedure, and an editorial policy checklist that verifies compliance with all required editorial policies

- <https://www.nature.com/documents/nr-reporting-summary.zi p>>Nature Research Reporting Summary
- <https://www.nature.com/documents/nr-editorial-policy-checklist.pdf>>Editorial Policy Checklist

All points on the policy checklist must be addressed. Your revised manuscript can only be sent back to the referees if these checklists are completed and uploaded with the revision.

Notes: If you have submitted a Stage 1 Registered Report, Review, Primer, Comment, or Perspective you do not need to submit these forms. If you have already submitted these forms, you may disregard this request.

* TRANSPARENT PEER REVIEW: Communications Psychology uses a transparent peer review system. This means that we publish the editorial decision letters including Reviewers' comments to the authors and the author rebuttal letters online as a supplementary peer review file. However, on author request, confidential information and data can be removed from the published reviewer reports and rebuttal letters prior to publication. If your manuscript has been previously reviewed at another journal, those Reviewers' comments would not form part of the published peer review file.

If you experience problems in linking your ORCID, please contact the <http://platformsupport.nature.com/> Platform Support Helpdesk.

Version 1:

Decision Letter: second round

Dear Dr Schafer,

Your manuscript titled "Social Avoidance as Multidimensional Navigation in Abstract Social Space" has now been seen by our reviewers, whose comments appear below. In light of their advice I am delighted to say that we are happy, in principle, to publish a suitably revised version in Communications Psychology.

We therefore invite you to revise your paper one last time to address the remaining concerns of our reviewers and a list of editorial requests. At the same time we ask that you edit your manuscript to comply with our format requirements and to maximise the accessibility and therefore the impact of your work.

EDITORIAL REQUESTS:

SUBMISSION INFORMATION:

In order to accept your paper, we require the files listed here <https://www.nature.com/documents/commsj-file-checklist.pdf> .

OPEN ACCESS:

* DATA AVAILABILITY:

All Communications Psychology manuscripts must include a section titled "Data Availability" at the end of the Methods section. More information on this policy, is available in the Editorial Requests Table and at <http://www.nature.com/authors/policies/data/data-availability-statements-data-citations.pdf> > <http://www.nature.com/authors/policies/data/data-availability-statements-data-citations.pdf> .

Link Redacted

Best regards,

Jennifer Bellingtier

Jennifer Bellingtier, PhD
Senior Editor
Communications Psychology

Patricia Lockwood, PhD
Editorial Board Member
Communications Psychology
orcid.org/0000-0001-7195-9559

REVIEWER EXPERTISE:

Reviewer #1 Social cognition, decision-making, navigation and cognitive maps

Reviewer #2 Social decision-making

REVIEWERS' COMMENTS:

Reviewer #1:

No further comments.

Reviewer #2 (Remarks to the Author):

I appreciate the author's effort in addressing my concerns. I have no further comments, and recommend this manuscript for publication.

We thank the reviewers for their comments on our manuscript, which have substantially improved the clarity and rigor of our work. Below, we respond point-by-point to each comment raised by the reviewers. For convenience, reviewer comments are put first in italics, organized by reviewer (with some grouping of related comments) followed by our responses and proposed revisions (in purple). Line numbers refer to the revised manuscript version.

Responses to reviewer 1

Comment 1.1

Reviewer

“This study utilizes many cutting-edge natural language processing (NLP) techniques. I believe it is important to report the potential limitations of these methods and consider supplementing them with additional methods or tools.”

Response

Thank you for this comment. We have addressed these concerns in the revised manuscript as follows.

We clarify that these analyses are intended to offer converging evidence across multiple assessments. The semantic embedding analysis is intended to offer converging evidence that participants subjectively represent the social locations from the task (lines 121-133):

To offer converging evidence for social location representations, we tested whether the Validation participants' open-ended descriptions of the characters reflect their affiliation and power locations. Using large language models, we embedded each participant's character-specific text free responses as semantic vectors, and then correlated the pairwise semantic distances with their pairwise behavioral location distances and pairwise self-reported placement distances. In other words, we asked whether characters with similar semantic representations also have close locations in the two-dimensional space. In two different language models, these correlations were significantly greater than 0 for both the behavioral (Validation sample: $W = 16664$, right-tailed $P = 0.011$; $W = 15573$, right-tailed $P = 0.03$) and the self-reported placement locations (Validation sample: $W = 23044$, right-tailed $P < 0.001$; $W = 19540$, right-tailed $P = 0.003$). This analysis offers converging evidence that participants' self-reported representations contain information about affiliation and power.

The sentiment analysis is intended to test the connection between social avoidance, social distance and negative emotion (lines 242-266):

We hypothesized that individuals with higher Social Avoidance scores would form more negative impressions of the characters, and that this would be reflected in both their behavior and text descriptions of the characters. Having already established that both affiliation and power independently relate to social avoidance, we used a social distance metric that combines the participant's average affiliation and power decisions, calculated as the length of the vector between the participant's point-of-view and character locations in the two-dimensional social space (see supplement sections **First-person social distance explains Social Avoidance** and **Social Avoidance is related to social distance above and beyond likability and relative impact of characters**). We tested whether participants with higher self-reported Social Avoidance and larger behavioral social distance report more negative impressions of the characters in their open-ended text descriptions (**hypothesis 3**).

We analyzed the sentiment expressed in the Validation sample's open-ended character descriptions using two different sentiment analysis models—a rules-based model and a large language model. Average sentiment related to social distance in both models, with larger distance relating to more negative emotion (rules-based model: $f\beta = -0.25$, $CI_{95\%} = [-0.37, -0.12]$, $t_{231} = -3.8$, left-tailed $P < 0.001$; language model: $f\beta = -0.29$, $CI_{95\%} = [-0.41, -0.16]$, $t_{231} = -4.5$, left-tailed $P < 0.001$). In a regression with all three self-report factors, average sentiment also specifically and negatively correlated with the Social Avoidance factor (rules-based model: $f\beta = -0.18$, $CI_{95\%} = [-0.31, -0.06]$, $t_{229} = -2.85$, left-tailed $P = 0.0047$; language model: $f\beta = -0.19$, $CI_{95\%} = [-0.32, -0.06]$, $t_{229} = -2.97$, left-tailed $P = 0.0017$; Mood and Compulsion non-significant). These free response sentiment results support our hypothesis: negative perceptions of others are related to social avoidance and its behavioral manifestation in relationships (see **figure 3**).

We also now add discussions about the limitations of each technique, and now include additional models for each analysis to ensure the results are robust across models. We note this in the methods section for the semantic embedding analysis (lines 779-785):

While these models are the cutting-edge of natural language processing, they can be sensitive to biases in their training data. To ensure our results do not depend on any given semantic space, we used two different models via the Hugging Face interface: all-MiniLM-L12-v2 and all-distilrobert-v1, both finetuned versions of published base models

(Sanh et al., 2019; Wang et al., 2020). Both models are pre-trained, bi-directional transformer neural networks trained to transform text into contextual embeddings that capture the semantic meaning of sentences.

We also note this for the character sentiment analysis (lines 830-838):

To test whether the Social Avoidance factor, social distancing behavior and negative social impressions are related, we conducted sentiment analysis on the Validation participants' free responses about the characters. As with the semantic embedding analysis, we tested this with two separate models to ensure robustness of the results. We used a widely used rule-based system named VADER (Hutto & Gilbert, 2014), which relies on a predefined lexicon and set of rules. We supplemented this model with a large language model finetuned to do sentiment analysis (Barbieri et al., 2020), which does not depend on the same rule-based assumptions and may be better suited to capture contextual nuances of language.

Comment 1.2

Reviewer

“Based on the introduction, I anticipated that the authors would analyze the impact of the social avoidance trait on trial-by-trial decisions during social interactions. However, the study design appears to measure decisions associated with low-affiliation and low-power tendencies after the social interactions, only once. This discrepancy between the introduction's expectations and the actual study design raises concerns about the validity of the hypothesis, findings, and interpretations. For example, the statement 'people highest in these symptoms should make low-affiliation and low-power decisions in the task's social interactions'; Similarly, the statement 'Here we sought to test the extent to which people high in self reported social avoidance show implicit low-affiliation and low-power behavioral tendencies in social interactions' needs to be re-evaluated considering the timing of the decision-making process.”

Response

We apologize for the confusion. To clarify, participants made multiple trial-by-trial decisions during their interactions with the characters. Each decision is an indication of an overall social tendency, implicit in the trial-by-trial choices. We analyze averages of these decisions (summarized in the average affiliation and power, and social distance), but the decision-making is during the simulated social interactions, not from post-hoc assessments.

The statements in the introduction reflect our study design: participants made social decisions throughout their interactions with each character. When we refer to “decisions during social interactions,” we mean these choices, which represents an active decision about how to interact with the character. While we aggregate the choices to assess overall behavioral patterns, these aggregates are calculated from multiple individual decisions made during the actual social interactions, not from post-hoc assessments. Thus, when we state that “people highest in these symptoms should make low-affiliation and low-power decisions in the task’s social interactions”, we are referring to these moment-to-moment choices that occurred during the task interactions.

After the task, we also assessed their subjective perception of the characters to see the correspondence of the subjective impressions to the implicit behavioral measures objectively in the game. This was to validate that participants subjectively represent the characters in terms of the behavioral locations from their choices.

In the revision, we carefully denote the measurements, clarifying the difference between the behavioral locations (from the task choices) and the self-report location placements (after the task). We have also added details to the task description in the methods, including a set of example choices, reproduced below (pages 21-22):

Decision dimension	Previous slide	Increase choice	Decrease choice
Affiliation	Chris goes in for a hug.	You hug him for a long moment.	You shake his hand instead.

Affiliation

Anthony is taking the elevator downstairs with you.

You: "Looks like you're the real person running this place - everybody really relies on you!"

You check your email on your phone: "Is it always that hectic in here?"

Affiliation	Mrs. Newcomb gets a phone call. Something is wrong; she seems very worried.	You feel concerned: "Everything alright, Jane? What is it?"	You don't want to get involved: "I'll give you privacy. I think we're done, anyway."
Power	Maya: "It doesn't matter if you're late. We're grabbing coffee."	You: "I can't - gotta go. Maybe tomorrow."	You really don't want to be late but reluctantly agree to grab coffee.
Power	Kayce is approaching the car. Chris gestures for you to move over to the middle seat.	You wait for Chris to take the middle seat.	You immediately move over to the middle seat.
Power	Anthony: "By the way, I have a lead on a place. It's a condo with two separate units."	You: "Get me the contract and floor plan and tell them you need 'til tomorrow."	You: "Let me know when you have more information."

Table 2. Examples of affiliation and power interaction decisions. Several examples of affiliation and power interactions are shown. The slide preceding the choice trial is shown to give the context for the interaction, along with the decision that would increase and the decision that would decrease the location along the relevant dimension.

Comment 1.3

Reviewer

“Could a single character's low affiliation or power reduce the average affiliation and power tendencies across all characters? It's unclear whether this would result in a general shift in the 2D social space, or if only certain characters within specific domains are more likely to be pulled towards lower values.”

Response

Thank you for this comment. We agree that this is an important consideration. In the initial submission, we performed separate regressions for each character to assess whether these effects were similar across characters. Indeed, all characters had negative associations between social avoidance factor and the affiliation/power tendencies, consistent with our main hypothesis. In the revision, we further tested for the possibility of character specific effects with mixed effects models. We now have a dedicated section in the supplement describing these analyses in detail (**testing for individual character effects**; lines 908-934):

To ensure that the effects of affiliation and power on Social Avoidance were not driven by specific characters, we tested for character-level effects using two complementary approaches.

First, we ran ordinary least squares regressions separately for each character, where the affiliation and power locations (along with standard controls) predicted Social Avoidance. The pattern was consistent with the overall effect across all characters, with both affiliation and power showing negative relationships with Social Avoidance (all β s < 0).

Second, we used linear mixed effects models with restricted maximum likelihood estimation. For affiliation and power separately, we constructed and compared nested models. The simpler model predicted Social Avoidance as a function of affiliation (or power), including the character identities and standard controls (affiliation ~ Social Avoidance + character identity + controls + (1| participant)). The more complex model

included additional interaction terms between character identities and affiliation (or power). Both models included random intercepts for participants (1|participant) to account for the multiple character values within each participant. Likelihood ratio tests comparing these models yielded non-significant results, for both affiliation and power (χ^2 tests, P s = 1), indicating that the effects of affiliation and power on Social Avoidance were consistent across characters. We conducted a similar analysis for social distance as the predicted variable. This was also non-significant (χ^2 test, $P > 0.6$), suggesting that the simpler model provided a better fit.

These analyses show that the relationships between our key variables—affiliation, power and social distance, and their associations with Social Avoidance—were consistent across the characters, rather than showing character-specific patterns.

Comment 1.4

Reviewer

“What is the range and variance of the responses for each clinical questionnaire? Can the authors compare these values with those from diagnosed populations? It is unclear to what extent the current participant sample is representative and whether the current data can be generalized to potential clinical populations.”

Response

We agree with the reviewer that more details regarding the clinical questionnaire responses would be useful. We now report the mean and standard deviation for each questionnaire in both samples in the supplement (**Questionnaire summary statistics**, pages 39-40):

Questionnaire (construct)	Initial sample mean (SD)	Validation sample mean (SD)
Zung Self-Rating Depression Scale (Depression)	38.3 (11)	41.3 (11.1)

Apathy Evaluation Scale (Apathy)	36.8 (12.7)	38.1 (12.4)
Liebowitz Social Anxiety Scale avoidance subscale (Social Anxiety)	32.4 (16.1)	34.8 (16)
Avoidant Personality Disorder Impairment Scale (Avoidant Personality)	9.9 (6.5)	10.2 (6.9)
Short Scales for Measuring Schizotypy (Schizotypy)	12.5 (7.7)	12.9 (7.8)
Broad Autism Phenotype Questionnaire (Autism)	3.1 (0.7)	3.2 (0.7)
Zanarini Borderline Personality Disorder (Borderline Personality)	0.32 (0.3)	0.33 (0.3)
Obsessive Compulsive Inventory-Revised (Compulsion)	12 (11.6)	16 (13.4)

As these questionnaires are not diagnostic questionnaires and our sample is derived from the general population with a low prevalence of self-reported psychiatric diagnoses, we highlight our first paragraph under the “Future work: clinical samples and social narratives” subsection of the discussion:

One strength of our approach is that we have two large samples that represent a broad swath of the general population. As we did not have any clinical assessments, and we had low prevalence of individuals with psychiatric diagnoses in the last 6 months, we were limited to learning from participants’ responses to self-report questionnaires. Future work should recruit clinical samples (e.g., with a DSM diagnosis of Social Anxiety

Disorder) to test for the same relationships we found here. Moreover, multiple disorders show social avoidance-like behaviors, and there are other social symptoms (e.g., communication deficits) that could be explored using this task.

Comment 1.5

Reviewer

“The manuscript’s explanation of the differences in explainability between the polar coordinate model (first-person point-of-view) and the Cartesian model is unclear. While the angle and Euclidean distance in the polar coordinate system theoretically contain the same information as the position on the affiliation and power axes in the Cartesian system (assuming a shared origin), the manuscript does not adequately explain how the polar coordinate values are computed or why this leads to different levels of explainability. To clarify the differences in explainability, the manuscript should provide a detailed explanation of how the angle and Euclidean distance are calculated in the polar coordinate model. This should include the specific formulas used and how they relate to the affiliation and power axes. Moreover, this will be appreciated if the authors can clearly articulate the specific reasons why the polar coordinate model might offer a different level of explainability compared to the Cartesian model.”

Response

We agree that this section is confusing. We address this in several ways.

We make the calculation of social distance and angle clearer in the methods section (lines 495-507):

We also represented these locations as the distances and angles (r , θ) of the locations relative to the theoretical, “first-person” point-of-view of the participant (by subtracting the maximum possible affiliation and neutral power location: (6,0); Tavares et al., 2015):

$$\text{social distance } (r) = \sqrt{(\text{affiliation}-6)^2 + \text{power}^2}$$

$$\text{social angle } (\theta) = \arccos(\text{power} / \sqrt{(\text{affiliation}-6)^2 + \text{power}^2})$$

The angle is in the range [0, 180°]. To capture the participants' overall social tendencies in the task, we calculated the means of these variables. We then z-scored the affiliation, power and distance values and transformed the angles with the cosine function. For more information on the first-person representation see the supplement (**First-person social distance explains Social Avoidance**).

We moved the first-person versus neutral origin analysis to the supplement, to simplify the flow of the argument. We clarify here that this first-person representation is justified theoretically and empirically (lines 936-983):

Social interactions may be represented from a first-person point-of-view, where individuals represent others' locations relative to themselves in social space (Trope & Liberman, 2010). Consistent with this, previous research using this social navigation task found that neural representations are better explained by such a first-person framework, where relationships are represented as distances and angles from the participant's perspective (Tavares et al., 2015). In this framework, the orientation of the social vector indicates the interaction between power and affiliation relative to oneself, while the length represents absolute social distance, with greater distances reflecting lower affiliation and larger power differences.

We compared the ability of two coordinate systems (neutral and first-person) to explain the Social Avoidance effects, by calculating distances and angles from a reference location for each affiliation and power location across the task. In the neutral coordinate system, these values were calculated from the origin coordinates (0, 0); angles were calculated as the counterclockwise angle between the vector from the origin and the positive affiliation axis:

$$\text{neutral social distance } (r) = \sqrt{\text{affiliation}^2 + \text{power}^2}$$

$$\text{neutral social angle } (\theta) = \arctan2(\text{power}, \text{affiliation})$$

The angles were converted to the range $[0, 360^\circ]$ and transformed with the sine and cosine functions.

For the first-person coordinate system, distances were calculated from the maximum affiliation value and neutral power value (6,0); angles were measured between participant-to-character vector and the positive power axis $[0, 180^\circ]$ (see Character relationships as affiliation and power trajectories in the methods for formulas). We normalized the distances by z-scoring and the angles by the cosine transformation.

Previous work suggested that the first-person reference frame explained neural data better than the neutral reference frame; to test whether this is also the case for self-reported Social Avoidance, we compared the ability of these models to explain Social Avoidance scores. OLS-based BIC scores confirmed the superiority of the first-person model relative to the neutral model in both samples (as judged by smaller BIC; Initial sample: 1744.28 vs. 1761.93; Validation sample: 818.08 vs. 824.89). Some of this may be explained by parsimony: in the neutral representation, distance, sine angle and cosine angle are needed to fully specify each location, but in the first-person representation, only distance and cosine angle are needed to specify each location.

We then tested the first-person social distance and angle coefficients to compare their ability to explain Social Avoidance. Distance had a consistently significant relationship to Social Avoidance across both samples (Initial: $f^2 = 0.21$, $CI_{95\%} = [0.13, 0.29]$, $t_{555} = 4.96$, right-tailed $P < 0.001$; Validation: $f^2 = 0.25$, $CI_{95\%} = [0.12, 0.37]$, $t_{230} = 3.84$, right-tailed $P < 0.001$), whereas angle did not (Initial: $\beta = 0.11$, $CI_{95\%} = [0.03, 0.19]$, $t_{555} = 2.62$, right-tailed $P = 0.0045$; Validation: $f^2 = 0.09$, $CI_{95\%} = [-0.04, 0.22]$, $t_{230} = 1.38$, right-tailed $P = 0.085$). These results suggest that while both social distance and social angle explain the Social Avoidance score, the social distance of the characters from the self is the best single predictor.

Comment 1.6

Reviewer

“The BIC values for the different models appear to be quite similar. Was the overall BIC calculated by summing the individual BIC values for each participant? If so, how many participants exhibited a lower BIC for the target model compared to the other models?”

Response

The overall BIC values were calculated for the full models rather than summing individual participant BICs, as our models were across subjects, not within subjects. In addition, it's the specificity of the pattern of BIC values, more so than the absolute differences, that we want to draw attention to: the lowest values across affiliation and power are for the social factor, in both samples.

These BIC value differences are considered meaningful, according to a common rule of thumb (Raferty et al., 1995): differences between 2-6 are considered positive evidence, differences between 6-10 are considered strong evidence, and >10 is considered very strong evidence. We have changed the text to explain this (lines 224-238):

To test for the specificity of these effects, we tested whether the Social Avoidance factor showed stronger effects with affiliation and power compared to the Mood and Compulsion factors. We ran OLS regressions for each combination of behavior (average affiliation and power) and factor (Social Avoidance, Mood, Compulsion), and compared the BIC scores, where positive BIC differences (LIBIC) indicate stronger evidence for Social Avoidance relative to the comparison factor. Across both samples, all LIBIC values favored Social Avoidance when compared to both Mood (Initial sample: affiliation LIBIC = 17.88, power LIBIC = 6.44; Validation sample: affiliation LIBIC = 9.74, power LIBIC = 2.51) and Compulsion (Initial sample: affiliation LIBIC = 20.4, power LIBIC = 6.34; Validation sample: affiliation LIBIC = 9.88, power LIBIC = 2.52) (see **figure 2**). Heuristically, these BIC differences are meaningful: a difference of 2 or more offers positive evidence in favor of the model with smaller BIC, with larger differences providing stronger evidence (Raftery, 1995). As such, our hypothesis of specific

relationships between affiliation and power and Social Avoidance was supported (**hypothesis 2**).

To make the differences between the self-report factors easier to understand, we also turned the table into panel B in the main results figure:

Comment 1.7 & 1.8

Reviewer

1.7 *“Please provide a Scree plot in the factor analysis and provide a more specific rationale behind the choice of three factors.”*

1.8 *“The readers would appreciate it if the authors could specify the questions corresponding to the X-axis, as well as the scales and scores on the Y-axis in Figure 3. Currently, it is unclear and less intuitive why the authors have arranged the factor scores in this particular manner within the cross-correlation graph. What information can be derived from the cross-correlation graph in Figure 3? Can the authors provide more details on the insights that readers should gain from this figure?”*

Response

We apologize for the confusion regarding the factor selection and figure.

Regarding the factor selection, we have edited the methods section to clarify our factor selection rationale (lines 648-654):

We retained factors based on Cattell’s criterion (Cattell, 1966). This approach sorts eigenvalues in descending order and identifies an “elbow”, a sharp drop-off in value where more factors may offer diminishing returns. We implemented this criterion using the Cattell-Nelson-Gorsuch test (Gorsuch & Nelson, 1981), which calculates the slopes for all possible sets of three adjacent eigenvalues and chooses the factors before where there is the greatest change in slope. This test showed that the difference in slopes is greatest at factor four, suggesting a three factor solution (see figure 5).

We also now include a Scree plot, reproduced below, within the consolidated factor analysis figure below.

Regarding the figure, we added it to show how the individual questionnaire items contribute to the different factors based on their correlation structure, which helps justify our factor labeling. Now, to improve the figure, we combined it with the word plot figure, added the screeplot and included the questionnaire names in the figure caption. We believe these improvements will help readers better understand the factor analysis.

Figure 5. Self-report factor analysis. (A) The scree plot shows the twenty largest eigenvalues, with the three factor solution highlighted in green. These three factors were selected based on when the eigenvalues start to flatten out. (B) Correlation matrix of the self-report questionnaire items (132 items across 8 questionnaires), with the 3 factor structure from the exploratory factor analysis shown on the axes (Initial sample: $n = 579$). The factors are labeled using the questionnaires with the highest average loadings. Questionnaire items are

color coded according to their questionnaire and questionnaires are labeled according to the symptom they measure. Avoidant Personality: Avoidant Personality Disorder Impairment Scale; Social Anxiety: Liebowitz Social Anxiety Scale avoidance subscale; Autism: Broad Autism Phenotype Questionnaire; Depression: Zung Self-Rating Depression Scale; Apathy: Apathy Evaluation Scale; Borderline Personality: Zanarini Borderline Personality Disorder; Schizotypy: Short Scales for Measuring Schizotypy; Compulsion: Obsessive Compulsive Inventory-Revised. (C) Word clouds showing the most frequent words from the top twenty most heavily weighted items for each self-report factor, with word size proportional to frequency.

We have also added a new table, that summarizes how the different questionnaires relate to each factor (page 31):

	Social Avoidance (factor 1)	Mood (factor 2)	Compulsion (factor 3)
Social Anxiety	0.55 (0.10)	0.06 (0.06)	0.17 (0.08)
Avoidant Personality	0.60 (0.10)	0.22 (0.06)	0.13 (0.09)
Autism	0.53 (0.17)	0.17 (0.10)	0.05 (0.15)
Apathy	0.28 (0.11)	0.58 (0.13)	-0.02 (0.08)
Depression	0.22 (0.09)	0.50 (0.17)	0.23 (0.13)
Borderline Personality	0.22 (0.07)	0.29 (0.08)	0.30 (0.02)
Obsessive Compulsive	0.13 (0.07)	0.12 (0.13)	0.54 (0.04)
Schizotypy	0.19 (0.17)	0.17 (0.09)	0.28 (0.14)

Table 1. Summary statistics (mean and standard deviations) of the loadings of different questionnaires onto the three factors. Average loadings greater than 0.25 are highlighted to emphasize the dominant questionnaires for each factor.

Responses to reviewer 2

Comments 2.2 & 2.6

Reviewer

2.6 “Currently, the theoretical background of the question at hand reads a bit weak and confusing. For example, the two main concepts (i.e., affiliation and power) are not clearly defined in the paper. The explanations provided “Affiliation decisions included whether to share physical touch, physical space, or information (e.g., to share their thoughts on a topic). Power decisions were whether to submit to or issue a directive/command, or otherwise exert or give control.” seem insufficient to me. Furthermore, based on the example given, it is unclear what “sharing physical touch or information” means (see also the comment below regarding the example included). This may generate confusion as (social) power definition and measurement, for instance, varies based on the scientific discipline (i.e., social psychology: influence the behavior, emotions, or beliefs of others; organizational psychology: capacity of leaders to influence the behavior of their followers to achieve organizational goals; experimental economics: control over the resources of another). The same applies for affiliation.”

2.2. "While I find the task and 2D mapping innovative, I am missing the crucial link. From my understanding of the paper, I am not convinced that the task really isolates features of social avoidance, but rather gets at the likeability of characters, beliefs about social norms and appropriate, or acceptable social behaviour in such contexts. I would like to see if, as mentioned above, the social navigation measures are not in fact correlated with other dimensions related to social interactions. Also, it is not clear for me from the presentation of the results, how connected are the questionnaires of social avoidance to the score in the task."

Response

We thank the reviewer for these comments. In the revised manuscript, we have addressed the concerns regarding the theoretical background, the definition of affiliation and power and alternative hypotheses in the following ways.

We changed the introduction to better emphasize connections to previous literature (lines 30-41):

Affiliation and power are fundamental in social behavior (Schafer & Schiller, 2018), underlying stereotyping (warmth and competence; Fiske, 2012), facial impressions (trustworthiness and dominance; Todorov et al., 2008), interpersonal traits and behaviors (Wiggins, 1979), as well as relationships of non-human primates (bonding and dominance; Feldblum et al., 2021) and other animals. These social dimensions manifest differently across people, with individuals high in social avoidance consistently self-reporting feeling low affiliation with others and feelings of powerlessness. For example, they report expecting social rejection (Downey & Feldman, 1996), behaving submissively (Allan & Gilbert, 1997), and lacking motivation to pursue affiliative interactions (Blay et al., 2021). However, some work suggests social avoidance is related to power but not affiliation (Dijk et al., 2018), and it is generally unclear if and how these self-reports relate to actual behaviors in everyday social settings.

Our task takes a naturalistic approach: each interaction decision has choices that vary primarily along either affiliation and power. As such, the operationalization of affiliation and power in a given situation is contextual, designed to reflect naturalistic variation in real-life interactions. To help clarify this, we now include more examples of affiliation and power decisions in the methods section (pages 21-22):

Decision dimension	Previous slide	Increase choice	Decrease choice
Affiliation	Chris goes in for a hug.	You hug him for a long moment.	You shake his hand instead.
Affiliation	Anthony is taking the elevator downstairs with you.	You: "Looks like you're the real person running this place - everybody	You check your email on your phone: "Is it always that hectic in here?"

really relies on
you!”

Affiliation	Mrs. Newcomb gets a phone call. Something is wrong; she seems very worried.	You feel concerned: “Everything alright, Jane? What is it?”	You don't want to get involved: “I'll give you privacy. I think we're done, anyway.”
Power	Maya: “It doesn't matter if you're late. We're grabbing coffee.”	You: “I can't - gotta go. Maybe tomorrow.”	You really don't want to be late but reluctantly agree to grab coffee.
Power	Kayce is approaching the car. Chris gestures for you to move over to the middle seat.	You wait for Chris to take the middle seat.	You immediately move over to the middle seat.
Power	Anthony: “By the way, I have a lead on a place. It's a condo with two separate units.”	You: “Get me the contract and floor plan and tell them you need 'til tomorrow.”	You: "Let me know when you have more information."

Table 2. Examples of affiliation and power interaction decisions. Several examples of affiliation and power interactions are shown. The slide preceding the choice trial is shown to give the context for the interaction, along with the decision that would increase and the decision that would decrease the location along the relevant dimension.

We agree with the reviewer on the importance of considering other social dimensions. In our Validation sample, we collected ratings of character likability from the participants (among other rating scales), as we wanted to better understand the meaning of the decisions. We expected these social dimensions to correlate with the affiliation and power decisions. As such, we don't view character likability (or impact) as being at odds with our hypotheses; in fact, we

think these judgments *reflect* our hypotheses. We have edited the description of these analyses in the results section to explain our hypothesis and interpretation of the results (lines 137-155):

Having established that participants represent the characters' social locations both explicitly (through social space placements) and implicitly (through open-ended text descriptions), we next sought to establish whether these dimensions map onto theoretically relevant social judgments. We predicted that affiliation choices should correlate with liking the characters, while power choices should correlate with the perceived relative impact of the characters on one's own goals. To test this, we had Validation participants rate how much they liked interacting with each character ("likability"), and how much control or influence they perceived each character to have relative to themselves ("relative impact"). Using regressions, we found a double dissociation: average character likability correlated with the average affiliation location ($f^2 = 0.40$, $CI_{95\%} = [0.29, 0.52]$, $t_{232} = 6.92$, right-tailed $P < 0.001$)—but not power ($P = 0.51$)—and average relative impact correlated with the average power location ($f^2 = 0.20$, $CI_{95\%} = [0.08, 0.33]$, $t_{232} = 3.18$, right-tailed $P < 0.001$)—but not affiliation ($P = 0.82$). Analyses of other social judgments confirmed the relative specificity of these relationships (see **self-reported character rating specificity analyses** in the supplement). Thus, the social interaction tendencies of the participants related to conceptually meaningful dimensions: participants who liked the characters made more affiliative decisions, and participants who rated themselves as having higher impact (i.e., influence on goals) than the characters made more dominant decisions.

To better understand how social avoidance relates to the task behavior, while controlling for character likability (and relative impact), we ran additional regressions (in the Validation sample only). We have included those results in the supplement (**Social Avoidance is related to social distance above and beyond likability and relative impact of characters**; lines 1025-1031):

To test whether the Social Avoidance factor was related to behavior above and beyond the effects of character likability and relative impact, we also regressed the Social Avoidance scores onto the social distance measure, average likability, average relative

impact as well as the standard controls. Even when controlling for these variables, social distance was significantly correlated with Social Avoidance ($\beta = 0.20$, $CI_{95\%} = [0.06, 0.33]$, $t_{230} = 3.84$, right-tailed $P = 0.0019$). This suggests that actual behavioral choices may be more closely linked to social avoidance than self-reported ratings.

We have added additional clarity to how the Social Avoidance score relates to the task behavior, in the text and in figures; for example the main results figure has a new panel clarifying the role of the BIC analysis, along with additional information in the caption:

Figure 2. Effects of self-report factors and task-based social relationship geometry. (A) As predicted, low affiliation (blue; more distance between participant and characters) and low power (red; less power to self, more power to characters) behaviors are related to high social avoidance-like symptoms. Affiliation and power behaviors

were included in each model, with controls (age, sex, race, IQ, presence of psychiatric diagnosis, task version, and task memory), predicting the different self-report factors. The bar heights are Ordinary Least Squares beta values (e.g., the effect related to average affiliation, controlling for average power and the controls), the error bars are 2-sided 95% confidence intervals and the asterisks are p-value significance (* < 0.05, ** < 0.01, *** < 0.005, **** < 0.001). Left-tailed p-values were used for the Social Avoidance factor as our hypotheses were directional. These effects were similar in both samples, and whether affiliation and power were modeled together or separately. (B) Social Avoidance had the smallest Bayesian Information Criterion (BIC) score for predicting the affiliation and power tendencies in both samples, suggesting the relationship between task behavior and self-reported factors was relatively specific to the Social Avoidance factor.

Comments 2.1 & 2.5

Reviewer

2.1: *“In the questionnaires used there are factors addressing challenging aspects related to social interactions, however, I think it would have been useful to also have some positive valenced self-reports in order to verify that there is no underlying aspect of social orientation, social skill, or social norm sensitivity that might affect the interpretation of the results.”*

2.5: *“It would improve the manuscript if a more solid justification (based on existing literature) regarding the rationale behind the choice of each of the batteries used to measure specific traits of the participants, and why this is limited to “negative” constructs. Explaining, also, why the choice to use specific dimensions and subscales of each battery for their hypotheses (maybe a pre-registration could have strengthened these choices). A better definition of the concepts referred to. The switch between whether the measured values were impressions, emotions or sentiments, as the authors do not clarify this sufficiently prior to presenting the results. More consistency in terminology could help the reader.”*

Response

We thank the reviewer for these thoughtful comments, which we respond to together, given their relatedness. In the revised manuscript (“Questionnaire selection” in Methods and Materials), we have clarified our rationale for selecting these specific questionnaires, as follows (lines 604-612):

Our questionnaire approach was inspired by (Gillan et al., 2016), and our questionnaire set overlapped with theirs: we collected the same Social Anxiety, Depression, Apathy, Compulsion and Schizotypy questionnaires. Previous research shows the fear and avoidance subscales of the Liebowitz Social Anxiety Scale (Social Anxiety) are highly correlated (e.g., Beard et al., 2011); thus, we only collected the avoidance subscale to reduce participant burden. To ensure coverage of social avoidance related items beyond Gillan et al.’s approach, we added the Avoidant Personality, Autism and Borderline Personality questionnaires.

We conceptualize our results in terms of social avoidance (a negative construct), but as our approach is dimensional, we can conceptualize them in terms of social approach. As such the social avoidance factor likely captures a general response tendency to approach or avoid social situations. For example, the sentiment analysis on the free responses about the characters suggests that more self-reported social avoidance and social distancing behavior both relate to negative impressions of the characters. But another way to phrase this is that less self-reported social avoidance and social distancing behavior relate to *positive* impressions of the characters. Consistent with this, the social avoidance factor strongly correlated with the real-world social network size. Moreover, some of the questionnaire items in the social avoidance factor questions are framed positively: for example, items include “I like being around people” and “I enjoy being in social situations.” We have added some text to clarify this (lines 614-622):

We focused on negative constructs given our interest in transdiagnostic symptoms associated with social avoidance, mood and compulsivity. However, these response scales capture general tendencies from positive to negative on these constructs, with some items being explicitly framed in positive terms.

In the revised discussion, we note that in future work we will consider other hypotheses (lines 379-381):

Future work should also consider alternative constructs, beyond self-reported social avoidance (e.g., sensitivity to social norms and context).

We also agree that more consistency in terminology will improve the manuscript; as such, we now use consistent terminology to describe the different analyses. We describe these and other related changes in our response to comments **2.3, 2.4 and 2.7**.

Comments 2.3, 2.4 & 2.7

Reviewer

2.3 *“All secondary analyses related to different aspects of the question at hand, could be labeled as “exploratory” and be included in the supplementary material. This would make the results less confusing and improve the flow of the manuscript.”*

2.4 *“If more standardized data visualizations related to each type of analyses and results were included, that could help. Most of the ones that are included seem to me non-intuitive in supporting findings. Too many different facets of findings are presented based on different types of analyses and plots. More consistency and clarity regarding the type of analysis and the chosen visualization for each hypothesis could help.”*

2.7 *“I believe that the amount of details in the methodology description could be sufficient for a replication and the statistical analysis is in order. However, it would be helpful to get a clearer justification why the specific analysis methods were chosen, and what each could add. Ideally, I would limit the included analyses per hypothesis presenting only statistical methods and associated results that answer the main question(s). The presented analysis seems to be more exploratory, or at least not immediately embedded in practices of the field. Moreover, it is not clearly connecting the methodology used to the RQs and the hypotheses. A more detailed explanation, and backing up of analysis methodology and results could help improve the paper.”*

Response

We thank the reviewer for these comments; addressing them has greatly improved the clarity of the manuscript. We have made significant changes to the organization and explanation in the manuscript, to account for all three comments, which we detail below.

We consolidated the descriptions of the questionnaire selection and factor analysis details in the Materials and Methods section. We also created a single figure (figure 4) to consolidate the figures about finding and labeling the self-report factors, as well as added a scree plot to visualize the selection of three factors, with additional information in the caption:

A

B

C

Figure 5. Self-report factor analysis. (A) The scree plot shows the twenty largest eigenvalues, with the three factor solution highlighted in green. These three factors were selected based on when the eigenvalues start to flatten out. (B) Correlation matrix of the self-report questionnaire items (132 items across 8 questionnaires), with the 3 factor structure from the exploratory factor analysis shown on the axes (Initial sample: $n = 579$). The factors are labeled using the questionnaires with the highest average loadings. Questionnaire items are color coded according to their questionnaire and questionnaires are labeled according to the symptom they measure. Avoidant Personality: Avoidant Personality Disorder Impairment Scale; Social Anxiety: Liebowitz Social Anxiety Scale avoidance subscale; Autism: Broad Autism Phenotype Questionnaire; Depression: Zung Self-Rating Depression Scale; Apathy: Apathy Evaluation Scale; Borderline Personality: Zanarini Borderline Personality Disorder; Schizotypy: Short Scales for Measuring Schizotypy; Compulsion: Obsessive Compulsive Inventory-Revised. (C) Word clouds showing the most frequent words from the top twenty most heavily weighted items for each self-report factor, with word size proportional to frequency.

We simplified, clarified and moved the first-person versus neutral origin analysis to the supplement, as this is a supporting analysis and one that confirms a previous finding using the same social navigation task.

Social interactions may be represented from a first-person point-of-view, where individuals represent others' locations relative to themselves in social space (Trope & Liberman, 2010). Consistent with this, previous research using this social navigation task found that neural representations are better explained by such a first-person framework, where relationships are represented as distances and angles from the participant's perspective (Tavares et al., 2015). In this framework, the orientation of the social vector indicates the interaction between power and affiliation relative to oneself, while the length represents absolute social distance, with greater distances reflecting lower affiliation and larger power differences.

We compared the ability of two coordinate systems (neutral and first-person) to explain the Social Avoidance effects, by calculating distances and angles from a reference location for each affiliation and power location across the task. In the neutral coordinate

system, these values were calculated from the origin coordinates (0, 0); angles were calculated as the counterclockwise angle between the vector from the origin and the positive affiliation axis:

$$\text{neutral social distance (r)} = \sqrt{\text{affiliation}^2 + \text{power}^2}$$

$$\text{neutral social angle } (\theta) = \arctan2(\text{power}, \text{affiliation})$$

The angles were converted to the range [0, 360°] and transformed with the sine and cosine functions.

For the first-person coordinate system, distances were calculated from the maximum affiliation value and neutral power value (6,0); angles were measured between participant-to-character vector and the positive power axis [0, 180°] (see **Character relationships as affiliation and power trajectories** in the methods for formulas). We normalized the distances by z-scoring and the angles by the cosine transformation.

Previous work suggested that the first-person reference frame explained neural data better than the neutral reference frame; to test whether this is also the case for self-reported Social Avoidance, we compared the ability of these models to explain Social Avoidance scores. OLS-based BIC scores confirmed the superiority of the first-person model relative to the neutral model in both samples (as judged by smaller BIC; Initial sample: 1744.28 vs. 1761.93; Validation sample: 818.08 vs. 824.89). Some of this may be explained by parsimony: in the neutral representation, distance, sine angle and cosine angle are needed to fully specify each location, but in the first-person representation, only distance and cosine angle are needed to specify each location.

We then tested the first-person social distance and angle coefficients to compare their ability to explain Social Avoidance. Distance had a consistently significant relationship to Social Avoidance across both samples (Initial: $f^2 = 0.21$, $CI_{95\%} = [0.13, 0.29]$, $t_{555} = 4.96$, right-tailed $P < 0.001$; Validation: $f^2 = 0.25$, $CI_{95\%} = [0.12, 0.37]$, $t_{230} = 3.84$, right-tailed $P < 0.001$), whereas angle did not (Initial: $\beta = 0.11$, $CI_{95\%} = [0.03, 0.19]$, $t_{555} = 2.62$, right-tailed $P = 0.0045$; Validation: $f^2 = 0.09$, $CI_{95\%} = [-0.04, 0.22]$, $t_{230} = 1.38$, right-

tailed $P = 0.085$). These results suggest that while both social distance and social angle explain the Social Avoidance score, the social distance of the characters from the self is the best single predictor.

We also removed the social quadrant bias analysis altogether, as this was meant mainly as a visual representation of the effect, but likely will just confuse the reader.

We have simplified the analysis terminology to make the results easier to follow. For the social navigation task behavior, we previously called the affiliation and power location averages “tendencies”; now, we have removed this as this introduces an unnecessary term, and simply call them affiliation and power averages, or (where the context is clear) simply affiliation and power.

We also now clarify the terminology and role for the three social navigation task validations. We reproduce the new results text below for each of these analyses, for convenience (lines 86-162):

Decision-making shaping social interactions is consistent with self-reported social mapping

The social navigation task aims to capture how people represent and navigate social relationships along two fundamental dimensions: affiliation and power. The task is a narrative-based game where participants make a series of decisions about how to interact with different characters in naturalistic social situations. Unbeknownst to the participants, each choice implicitly moves the character along either an affiliation or power dimension in abstract social space, creating social trajectories (see **figure 1**). We tested whether these behavioral locations reflect subjective social mapping by analyzing three different post-task measures: self-reported placements of the characters onto an affiliation and power space to validate the location representation, free response semantic representations of the characters for convergent evidence of location

representation, and self-reported ratings of character liking and relative impact to validate the social dimensions themselves.

Behavioral affiliation and power locations relate to self-reported character location placements

To test if the participants represent the social locations implied by their behavioral choices, we asked whether self-reported (i.e., explicit) placement of characters in social space align with the behavioral (i.e., implicit) locations from task decision (see **figure 1**). After the social navigation task, we had participants place the characters onto an affiliation and power space according to their own impressions of the characters (this was the first time they became aware of these dimensions), and then tested if the behavioral and self-reported locations were correlated. The behavioral and self-reported location placements were on average closer in social space than chance (Initial sample: $t_{578} = -18.2$, $P < 0.001$; Validation sample: $t_{254} = -10.61$, $P < 0.001$). Representational similarity analysis showed that the patterns of distances between characters were also similar between the behavioral and self-reported locations (Initial sample: $W = 97610$, $P < 0.001$; Validation sample: $W = 22072$, $P < 0.001$). These results suggest participants represent these relationships along the affiliation and power axes, despite the dimensions never being explicitly mentioned in the pre-task instructions or narrative.

Behavioral and self-reported affiliation and power locations relate to verbal descriptions of the characters

To offer converging evidence for social location representations, we tested whether the Validation participants' open-ended descriptions of the characters reflect their affiliation and power locations. Using large language models, we embedded each participant's character-specific text free responses as semantic vectors, and then correlated the pairwise semantic distances with their pairwise behavioral location distances and pairwise self-reported placement distances. In other words, we asked whether characters with similar semantic representations also have close locations in the two-dimensional space. In two different language models, these correlations were

significantly greater than 0 for both the behavioral (Validation sample: $W = 16664$, right-tailed $P = 0.011$; $W = 15573$, right-tailed $P = 0.03$) and the self-reported placement locations (Validation sample: $W = 23044$, right-tailed $P < 0.001$; $W = 19540$, right-tailed $P = 0.003$). This analysis offers converging evidence that participants' self-reported representations contain information about affiliation and power.

Behavioral affiliation and power dimensions relate to self-reported ratings of character liking and impact

Having established that participants represent the characters' social locations both explicitly (through social space placements) and implicitly (through open-ended text descriptions), we next sought to establish whether these dimensions map onto theoretically relevant social judgments. We predicted that affiliation choices should correlate with liking the characters, while power choices should correlate with the perceived relative impact of the characters on one's own goals. To test this, we had Validation participants rate how much they liked interacting with each character ("likability"), and how much control or influence they perceived each character to have relative to themselves ("relative impact"). Using regressions, we found a double dissociation: average character likability correlated with the average affiliation location ($f^2 = 0.40$, $CI_{95\%} = [0.29, 0.52]$, $t_{232} = 6.92$, right-tailed $P < 0.001$)—but not power ($P = 0.51$)—and average relative impact correlated with the average power location ($f^2 = 0.20$, $CI_{95\%} = [0.08, 0.33]$, $t_{232} = 3.18$, right-tailed $P < 0.001$)—but not affiliation ($P = 0.82$). Analyses of other social judgments confirmed the relative specificity of these relationships (see **self-reported character rating specificity analyses** in the supplement). Thus, the social interaction tendencies of the participants related to conceptually meaningful dimensions: participants who liked the characters made more affiliative decisions, and participants who rated themselves as having higher impact (i.e., influence on goals) than the characters made more dominant decisions.

Together, these analyses provide converging evidence that the social navigation task captures how people represent and navigate relationships along the axes of affiliation and power. Participants' affiliation and power choices were evident in their explicit spatial mappings, open-ended text descriptions, and social judgments of the

characters—despite the underlying affiliation and power dimensions never being explicitly mentioned in the task.

We now clarify that the factor that we labeled “social avoidance” is called either self-reported social avoidance or the Social Avoidance factor. We also clarify the free response sentiment analysis, and that it is a test of the Social Avoidance effect, as below (lines 830-832):

To test whether the Social Avoidance factor, social distancing behavior and negative social impressions are related, we conducted sentiment analysis on the Validation participants' free responses about the characters.

And here (lines 255-266):

We analyzed the sentiment expressed in the Validation sample's open-ended character descriptions using two different sentiment analysis models—a rules-based model and a large language model. Average sentiment related to social distance in both models, with larger distance relating to more negative emotion (rules-based model: $f_3 = -0.25$, $CI_{95\%} = [-0.37, -0.12]$, $t_{231} = -3.8$, left-tailed $P < 0.001$; language model: $f_3 = -0.29$, $CI_{95\%} = [-0.41, -0.16]$, $t_{231} = -4.5$, left-tailed $P < 0.001$). In a regression with all three self-report factors, average sentiment also specifically and negatively correlated with the Social Avoidance factor (rules-based model: $f_3 = -0.18$, $CI_{95\%} = [-0.31, -0.06]$, $t_{229} = -2.85$, left-tailed $P = 0.0047$; language model: $f_3 = -0.19$, $CI_{95\%} = [-0.32, -0.06]$, $t_{229} = -2.97$, left-tailed $P = 0.0017$; Mood and Compulsion non-significant). These free response sentiment results support our hypothesis: negative perceptions of others are related to social avoidance and its behavioral manifestation in relationships (see **figure 3**).

Finally, we also ensure the phrase “real-world” is used to describe the social network effect, to clarify this is a measure of real-world social networks, not any measure about the social navigation task.